

# Impacts of aerosol direct effects on tropospheric ozone through changes in atmospheric dynamics and photolysis rates

Jia Xing[1], Jiandong Wang[1], Rohit Mathur[2], Shuxiao Wang[1], Golam Sarwar[2], Jonathan Pleim[2], Christian

Hogrefe[2], Yuqiang Zhang[2], Jingkun Jiang[1], David C. Wong[2], Jiming Hao[1]

[1] State Key Joint Laboratory of Environmental Simulation and Pollution Control, School of Environment, Tsinghua University, Beijing 100084, China
[2] The U.S. Environmental Protection Agency, Research Triangle Park, NC 27711, USA

*These authors contributed equally to this work: Jia Xing & Jiandong Wang*

*Correspondence to: Shuxiao Wang (email: shxwang@tsinghua.edu.cn; phone: +86-10-62771466; fax: +86-10-62773650)*

**Abstract.** Aerosol direct effects (ADE), i.e., scattering and absorption of incoming solar radiation, reduce radiation reaching the ground and the resultant photolysis attenuation can decrease ozone ($O_3$) formation in polluted areas. One the other hand, evidence also suggests that ADE associated cooling suppresses atmospheric ventilation thereby enhancing surface-level $O_3$. Assessment of ADE impacts is thus important for understanding emission reduction strategies that seek co-benefits associated

with reductions in both particulate matter and $O_3$ levels. This study quantifies the impacts of ADE on tropospheric ozone by using a two-way online coupled meteorology and atmospheric chemistry model, WRF-CMAQ, instrumented with process analysis methodology. Two manifestations of ADE impacts on $O_3$ including changes in atmospheric dynamics (ΔDynamics) and changes in photolysis rates (ΔPhotolysis) were assessed separately through multiple scenario simulations for January and July of 2013 over China. Results suggest that ADE reduced surface daily maxima 1h $O_3$ (DM1O$_3$) in China by up to 39 μg m⁻

³ through the combination of ΔDynamics and ΔPhotolysis in January, but enhanced surface DM1O$_3$ by up to 4 μg m⁻³ in July. Increased $O_3$ in July is largely attributed to ΔDynamics which causes a weaker $O_3$ sink of dry deposition and a stronger $O_3$ source of photochemistry due to the stabilization of atmosphere. Meanwhile, surface OH is also enhanced at noon in July, though its daytime average values are reduced in January. An increased OH chain length and a shift towards more VOC-limited condition are found due to ADE in both January and July. This study suggests that reducing ADE may have potential

risk of increasing $O_3$ in winter, but it will benefit the reduction of maxima $O_3$ in summer.

## 1. Introduction

Photochemistry in the atmosphere is a well-known source for tropospheric ozone ($O_3$) (e.g., Haagen-Smit and Fox, 1954) and is determined by ambient levels of $O_3$ precursors (i.e., $NO_x$ and VOC) and photolysis rates which are largely influenced by meteorological factors such as solar irradiance and temperature. It is well known that aerosols influence radiation through light





scattering and absorption, thereby modulating atmospheric radiation and temperature. These aerosol direct effects (ADE) can then impact thermal and photochemical reactions leading to formation of $O_3$ (Dickerson et al., 1997). Recent studies suggest that the aerosol induced reduction in solar irradiance leads to lower photolysis rates and less $O_3$ (e.g., Benas et al., 2007), therefore extensive aerosol reductions, particularly in developing regions such as in East Asia, may pose a potential risk by

enhancing $O_3$ levels (Bian et al., 2007; Anger et al., 2016; Wang et al., 2016). For example, Wang et al (2016) found that because of ADE, the surface 1h maximum ozone (noted as $DM1O_3$) was reduced by up to 12% in eastern China during the EAST-AIRE campaign, suggesting that benefits of $PM_{2.5}$ reductions may be partially offset by increases in ozone associated with reducing ADE.

Ambient $O_3$ levels are influenced by several sources and sinks. The modulation of photolysis rates by ADE is only one

manifestation of ADE impacts on $O_3$. In addition, ADE modulate the temperature, atmospheric ventilation, cloud and rainfall which also influence the $O_3$ concentrations. Therefore, ADE can impact air quality through multiple pathways and process chains (Jacobson, 2002; 2010; Jacobson et al., 2007; Wang et al., 2014; Xing et al., 2015a; Ding et al., 2016). For example, Xing et al (2015a) suggested that the $O_3$ response to ADE is largely contributed by the increased precursor concentrations which enhance the photochemical reaction, presenting an overall positive response of $O_3$ to ADE by up to 2-3% in eastern

China. Assessment of separate contribution from individual processes is necessary for fully understanding how ADE impact $O_3$.

In China, atmospheric haze is currently one of the most serious environmental issue of concern. Over the next decade, the national government plans to implement stringent control actions aimed at lowering the $PM_{2.5}$ concentrations. Speculation on whether such extensive aerosol controls will enhance $O_3$ and oxidation capacity need to be carefully assessed and quantified.

Accurate assessment of the multiple ADE impacts is a prerequisite for accurate policy decision. The process analysis (PA) methodology is an advanced probing tool that enables quantitative assessment of integrated rates of key processes and reactions simulated in the atmospheric model (Jang et al, 1995; Zhang et al., 2009; Xu et al., 2008; Liu et al., 2010; Xing et al., 2011). In this study, we apply the PA methodology in the two-way coupled meteorology and atmospheric chemistry model, i.e., Weather Research and Forecast (WRF) model coupled with the Community Multiscale Air Quality (CMAQ) model developed

by U.S. Environmental Protection Agency (Pleim et al., 2008; Mathur et al., 2010; Wong et al., 2012; Yu et al., 2013; Mathur et al., 2014; Xing et al., 2015b), to examine the process chain interactions arising from ADE and quantify their impacts on $O_3$ concentration.

The manuscript is organized as following. A brief description of the model configuration, scenario design and PA method is presented in section 2. The $O_3$ response to ADE is discussed in section 3.1. PA analyses are discussed in section 3.2-3.3. The

summary and conclusion is provided in section 4.



## 2. Method

### 2.1 Modeling System

The two-way coupled WRF-CMAQ model has been detailed and fully evaluated in our pervious papers (Wang et al., 2014; Xing et al., 2015a, b). In the model version used here, concentrations of gaseous species and primary and secondary aerosols

are simulated by using Carbon Bond 05 gas-phase chemistry (Sarwar et al., 2008) and AERO6 aerosol module (Appel et al., 2013). The aerosol optical properties were estimated by the BHCOAT coated-sphere module (Bohren and Huffman, 1983) based on simulated aerosol composition and size distribution (Gan et al., 2015). In the coupled model, the estimated aerosol optical properties are fed to the RRTMG radiation module in WRF, thus updating the simulated atmospheric dynamics which then impact the simulated temperature, photolysis rate, transport, dispersion, deposition and cloud mixing and removal of

pollutants. Due to large uncertainties associated with the representation of aerosol impacts on cloud droplet number and optical thickness, the indirect radiative effects of aerosols are not included in the current calculation.

The gridded emission inventory, initial and boundary conditions are consistent with our previous studies (Zhao et al., 2013a, b; Wang et al., 2014), while the simulated domain is extended slightly to cover the entire China, as shown in Figure 1. A better model performance in the simulation of dynamic fields including total solar radiation, PBL height data as well as $PM_{2.5}$

concentrations were suggested after the inclusion of ADE (Wang et al., 2014). In this study, the model performance in the simulation of $O_3$ will be evaluated through the comparison with observations from 74 cities across China from the China National Urban Air Quality Real-time Publishing Platform (http://113.108.142.147:20035/emcpublish/). The simulation period is selected as January $1^{st}$ to $31^{st}$ and July $1^{st}$ to $31^{st}$ in 2013 to represent winter and summer conditions, respectively. Five regions are selected for analysis, including Jing-Jin-Ji area (denoted JJJ), Yangzi-River-Delta (denoted YRD), Perl-River-

Delta (denoted PRD), Sichuan Basin (denoted SCH) and Hubei-Hunan area (denoted HUZ), as shown in Figure 1.

### 2.2 Simulation Design

Table 1 summarizes the scenario design in this study. In the baseline simulation (denoted SimBL), no aerosol feedbacks either on photolysis rates or radiations was taken into account. In simulation SimNF, only aerosol feedbacks on photolysis rates were considered by embedding an inline photolysis calculation in the model which accounted for modulation of photolysis due to

ADE. Finally, in simulation SimSF aerosol feedbacks were considered on both photolysis rates and radiation calculations. Differences between the simulations of SimNF and SimBL are considered as ADE impacts on $O_3$ through photolysis (denoted ΔPhotolysis). Similarly, differences between the simulations of SimSF and SimNF are considered as the ADE impacts on $O_3$ through dynamics (denoted ΔDynamics), and differences between the simulations of SimSF and SimBL represents as the combined ADE impacts on $O_3$ due to both photolysis and dynamics (denoted ΔTotal).



### 2.3 Process Analysis

In this study the PA methodology is used in the WRF-CMAQ model to analyze processes impacting simulated $O_3$ level. The Integrated Process Rates (IPRs) track hourly contributions to $O_3$ from seven major modeled atmospheric processes that act as sinks or sources of $O_3$. These processes are gas phase chemistry (denoted CHEM), cloud processes (i.e., the net effect of

aqueous-phase chemistry, below- and in-cloud scavenging, and wet deposition, denoted CLDS), dry deposition (denoted DDEP), horizontal advection (denoted HADV), horizontal diffusion (denoted HDIF), vertical advection (denoted ZADV), and turbulent mixing (denoted VDIF). The difference in IPRs among SimBL, SimNF and SimSF represents the response of individual process to ADE. To enable the consistent examination of changes in the process due the ADE across all concentration ranges, we examine changes in the IPRs normalized by the $O_3$ concentrations. The differences in these process

rates (expressed in units of $hr^{-1}$) between the SimBL, SimSF, SimNF then provide estimates of the changes in process rates resulting from ADE and are shown in the $2^{nd}$-$4^{th}$ columns of Figure 4, and (b)-(d) of Figure 5 and 6.

Integrated Reaction Rates (IRRs) are used to investigate the relative importance of various gas-phase reactions in $O_3$ formation. Following the grouping approach of previous studies (Zhang et al., 2009; Liu et al., 2010; Xing et al., 2011), the chemical production of total odd oxygen ($O_x$) and the chain length of hydroxyl radical (OH) are calculated. Additionally, the ratio of the

chemical production rate of $H_2O_2$ to that of $HNO_3$ ($P_{H2O2}/P_{HNO3}$) is an estimated indicator of $NO_x$- or VOC- limited conditions for $O_3$ chemistry.

### 3. Results

### 3.1 O3 response to ADE

The simulated surface $DM1O_3$ in SimBL, SimNF and SimSF are compared in Figure 2a-c. In January, higher $DM1O_3$

concentrations are seen in southern China (SCH) where solar radiation is stronger than in the north. The model generally captured the spatial pattern with highest $DM1O_3$ in SCH over the simulated domain. In July, high $DM1O_3$ areas are located towards the north, especially in the JJJ and YRD regions which have relatively larger NOx and VOC emission density and favorable meteorological condition (e.g., less rain and moderate solar radiation).

In January, $O_3$ production in north China is VOC-limited regime, thus increase in $NO_x$ at surface stemming from the stabilized

atmosphere by ADE inhibits $O_3$ formation due to enhanced titration by NO. As seen in Figure 2d, the ΔDynamics reduced $DM1O_3$ in eastern China by up to 24 μg m$^{-3}$, but slightly increased $DM1O_3$ in parts of southern China by up to 7 μg m$^{-3}$. The decrease in incoming solar radiation due to ADE significantly reduces the photolysis rates in east China. As seen in Figure 2e, the ΔPhotolysis reduced $DM1O_3$ domain-wide by up to 16 μg m$^{-3}$. The combined effect of both ΔDynamics and ΔPhotolysis, results in an overall reduction in $DM1O_3$ as evident across the JJJ and SCH regions with monthly-average reductions up to 39

μg m$^{-3}$.



In July, the $O_3$ chemistry changes from a VOC-limited to a $NO_x$-limited regime across most of China. Therefore, increase in $NO_x$ concentration due to the stabilization of atmosphere associated with the ADE, facilitates $O_3$ formation. The $\Delta$Dynamics increased $DM1O_3$ across most areas of China, particularly in JJJ, YRD and SCH by up to 5 $\mu g\ m^{-3}$, with the exception of the PRD region where $DM1O_3$ decreased. The $\Delta$Photolysis results in contrasting impacts in July compared to January, as it

increased $DM1O_3$ in most polluted areas including JJJ, YRD, PRD, HUZ, although the solar radiances were reduced due to $\Delta$Photolysis. This behavior is likely due to enhanced aerosol scattering associated with higher summer-time $SO_4^{2-}$ levels during summer (He and Carmichael, 1999; Jacobson, 1998). The resultant enhancements in photolysis rates can then cause the noted higher concentrations. More importantly, the diurnal analysis (discussed in the next section) suggested that the reduced photolysis during the early morning in SimNF, enhances the ambient precursor concentrations (due to less reaction in early

morning) at noon when $O_3$ reaches the daily maximum. This increase in precursor concentrations then leads to enhanced $O_3$ formation later in the day which compensates for or even overwhelms the disbenefit from the reduced photolysis rate. In summer, $\Delta$Dynamics results in a much stronger influence on $DM1O_3$ than $\Delta$Photolysis, and the combined impact of ADE increased $O_3$ in most of regions in China by up to 4 $\mu g\ m^{-3}$.

The impact of the ADE on $O_3$ is further explored by examining the relationship between the observed and simulated $O_3$

concentrations ($DM1O_3$, daily values of the cities located in each region) as a function of the observed $PM_{2.5}$ concentrations (observed daily averaged values in those cities), as displayed in Figure 3. In regards to model performance for $DM1O_3$ simulations, generally, the model exhibits slightly high bias in January but low bias in July across the 5 regions. The inclusion of ADE moderately reduced $O_3$ concentration in January and slightly increased $O_3$ in July, resulting in reduction in bias and improved performance for $DM1O_3$ simulation in both January and July.

Interestingly, in most regions (expect JJJ in January), higher $O_3$ concentration occur with higher $PM_{2.5}$ concentrations, which is evident in both observations and simulations, suggestive of common precursors (e.g., $NO_x$), source sectors, and/or transport pathways contributing to both $O_3$ and $PM_{2.5}$ in these regions. In JJJ, however, where ADE is the strongest among the regions (see Figure 2), a negative correlation between $O_3$ and $PM_{2.5}$ is evident in January when the $PM_{2.5}$ can reach levels as high as 700 ug $m^{-3}$, indicating the strong ADE impacts on $O_3$ through both feedbacks to dynamics and photolysis which significantly

reduced $O_3$.

### 3.2 IPRs response to ADE

To further explore the ADE impacts on simulated $O_3$, the integrated process contributions are further analyzed in three ways: (a) 24-hour diurnal variations of process contributions to simulated surface $O_3$ (Figure 4), (b) vertical profiles from ground up to 1357 m AGL (above ground level, in model layer 1-10) during three key periods of the day (early morning, noon and late

afternoon) (Figure 5), and (c) correlations with near-ground $PM_{2.5}$ (average concentrations between the ground and 355m AGL, model layer 1-5) (Figure 6). In the following, we limit our discussion to analysis of model results for the JJJ region which





exhibited the strongest ADE among the regions; similar results were found for the other 4 regions and can be found in the Supporting Information section.

Diurnal variation of process contributions from chemistry (CHEM), dry deposition (DDEP) and vertical turbulent mixing (VDIF) which together contribute to more than 90% of the $O_3$ rate of change for the JJJ region, are illustrated in Figure 4. The

diurnal variation of IPRs for other processes and their response to ADE are displayed in Figure S1 for JJJ and Figure S2-S5 for other 4 regions.

For surface-level $O_3$, VDIF is the major source and DDEP is the major sink (Figure S1). The stabilization of atmosphere due to $\Delta$Dynamics leads to lower dry deposition rates (due to lower dry deposition velocity) and thus increases surface $O_3$. The largest impact of $\Delta$Dynamics on DDEP occurs during early morning and late afternoon which is consistent with the response

of the PBL height to ADE noted in our previous analysis (Xing et al., 2015).

Expectedly, CHEM is the second largest sink for surface $O_3$ during January, but a source for surface $O_3$ during the daytime in July. The $\Delta$Dynamics increased the surface $O_3$ around noon in both January and July for almost all regions (no impacts in PRD and YRD in January, see Figure S2-S3), since increased stability due to $\Delta$Dynamics concentrated more precursors locally, leading to enhanced $O_3$ formation during the photochemically most active period of the day. The $\Delta$Dynamics reduced the

surface $O_3$ around late afternoon in January at all regions. This is because the increased atmospheric stability during late afternoon and evening hours increased $NO_x$ concentration which titrated more $O_3$. The $\Delta$Photolysis reduced surface $O_3$ in all regions in January. These reductions were more pronounced during the early morning hours when the photolysis rate are most sensitive to the radiation intensity. The $\Delta$Photolysis resulted in comparatively larger reductions in surface $O_3$ during the early morning and late afternoon hours in July, but slightly increased surface $O_3$ at noon for most of the regions. This increase in $O_3$

can be hypothesized to result from the following sequence of events. Slower photochemical reaction in the morning in the $\Delta$Photolysis case lead to higher levels of precursors, whose accumulation then enhances $O_3$ formation at noon. This hypothesis is further confirmed by the changes in the diurnal variation of $NO_2$ which suggest that higher NO to $NO_2$ conversion during early morning results in enhanced daytime $NO_2$ levels (see Figure S6), consequently leading to higher noon-time $O_3$.

For aloft $O_3$ (from 100 to 1600 meters above ground) as seen in Figure 5, CHEM is the major source for $O_3$ at noon both in

January and in July. However, during the morning and afternoon hours, CHEM is a major source for $O_3$ in July, but a major sink in January. At noon in both January and July, the $\Delta$Dynamics increased near-surface $O_3$ (below 500m, model layer 1-6), but reduced upper-level $O_3$ (above 500m, model layer 7-10), because increased stability of the atmosphere concentrated precursors emissions within a shallower layer resulting in higher $O_3$ production. The $\Delta$Dynamics also reduced the near-surface $O_3$ during morning and afternoon in January. This might be due to VOC-limited chemistry during morning and late afternoon

hours, so that increased $NO_x$ concentrations result in greater $O_3$ titration. The $\Delta$Photolysis case considerably reduced upper-level $O_3$ in January. In July, $\Delta$Photolysis reduced upper-level $O_3$ in the morning and afternoon, but increased $O_3$ at noon. Higher





levels of precursors at noon might be the reason for such enhancement (see Figure S6).

The daytime near-ground-averaged (between the ground and 350m AGL, layers 1-5) IPR responses to ADE are shown in Figure 6 for JJJ and in Figure S7 for other regions. The IPR and its responses are presented as a function of near-ground-averaged $PM_{2.5}$ concentrations. As shown in Figure 6, as $PM_{2.5}$ concentrations increase, the positive contribution of CHEM in

July become larger while the negative contribution of CHEM in January become smaller. The ΔDynamics enhanced CHEM and thus increased $O_3$ concentration in both January and July, and such enhancement are generally larger for higher $PM_{2.5}$ loading. In contrast, in January ΔPhotolysis resulted in higher rates of $O_3$ destruction due to chemistry (negative contribution of CHEM), and the magnitude of this sink increased as $PM_{2.5}$ concentrations increase. The reduction of $O_3$ stemming from the enhancements in the chemical sinks due to ΔPhotolysis is the dominant impact of ADE in January. The enhanced positive

contribution of CHEM due to ΔDynamics was partially compensated by the reduction from ΔPhotolysis, resulting in a slight increase in the positive CHEM contribution to $O_3$ in July.

DDEP is the major sink of daytime $O_3$ during both January and July. The increased stability due to ΔDynamics reduced deposition velocity and thus increases $O_3$. These effects become larger with increasing $PM_{2.5}$ concentrations. The ΔPhotolysis has almost no impacts on DDEP. Thus, weaker removal of $O_3$ from DDEP associated with ADE, contributed to higher $O_3$ in

most regions during both January and July. Enhanced $O_3$ source of CHEM and reduced $O_3$ sink of DDEP stemming from ΔDynamics is the dominant impact of ADE in July.

### 3.3 IRR response to ADE

The simulated mid-day average (11:00-13:00 local time) surface $O_x$ (defined as the sum of O, $O_3$, $NO_2$, $NO_3$, $N_2O_5$, $HNO_3$, PNA, NTR, PAN and PANX) and OH and their responses to ADE are shown in Figure 7. Both $O_x$ and OH are significantly

reduced in the ΔPhotolysis case in January throughout the modeling domain. Both $O_x$ and OH also show reductions in the middle portions of east China in the ΔDynamics case in January. Together, the combined ADE impacts result in reduced $O_x$ and OH in January, with widespread reductions primarily due to ADE on photolysis. In July, ΔPhotolysis increased mid-day OH across most of China (Figure 7) which is consistent with the increase of $O_3$ at noon stemming from a higher level of precursors accumulation due to ΔPhotolysis. The overall ADE impact on OH is controlled by ΔPhotolysis, and result in

increased mid-day OH across most of China. For $O_x$, however, the impact of ΔDynamics overwhelms the impact from ΔPhotolysis, resulting in increase in $O_x$ concentrations in east China including YRD, SCH and HUZ.

To further examine the response of $O_x$ to ADE, in Figure 8 we examine vertical profiles of the integrated reaction rates at noon for the JJJ region. The stabilization of the atmosphere due to ΔDynamics concentrates precursors within a lower PBL, resulting in an increased total $O_x$ production rate ($P_{totalOx}$) mostly in near-ground model layers (below 500m, model layer 1-6); in

magnitude aloft (above 500m, model layer 7-10), this change in $P_{totalOx}$ is smaller in January, and become decreasing in July. The reduction of $P_{totalOx}$ due to ΔPhotolysis is greatest at the surface in January, and declines with altitude, and even becomes





reversed at high layers (about 1300m, model layer 10) (Figure 8a). The overall ADE impact in January is mainly dominated by ΔPhotolysis which largely overwhelms the impact of ΔDynamics (Figure 8a). However, in July, ΔPhotolysis enhanced $P_{totalOx}$ across all layers. The $P_{totalOx}$ shows small decreases at high altitudes but significant increase in near-ground model layers (below 500m, model layer 1-6) due to the combined ADE in July.

The changes in vertical profiles of production rates of new OH ($P_{NewOH}$) and reacted OH ($P_{ReactedOH}$) are similar to those of $P_{totalOx}$, with the noted decreases in January dominated by ΔPhotolysis. In contrast, the increases in July result from contribution from both ΔPhotolysis and ΔDynamics.

Analysis of the chain length is important to understand the characteristics of chain reaction mechanisms. The OH chain length (denoted OH_CL) is determined by the ratio of $P_{ReactedOH}$ to $P_{NewOH}$. ΔDynamics concentrated more $NO_x$ at surface, thus leading

to an increased OH_CL (i.e., more reacted OH than new OH) in the near-ground layers, but a decreased OH_CL in the upper layers. In January, the ΔPhotolysis reduced $P_{NewOH}$ more than $P_{ReactedOH}$ (probably because of more abundance of $NO_x$ resulting from photolysis attenuation and consequently reduced photochemistry), thereby leading to an increased OH_CL. In July, ΔPhotolysis enhanced both $P_{NewOH}$ and $P_{ReactedOH}$, particularly in the upper layers. The OH_CL is increased by ΔPhotolysis because higher $NO_x$ levels (see Figure S6) cause more reacted OH to be reacted. Thus the surface OH_CL at noon is increased

in both January and July from combined ADE of ΔPhotolysis and ΔDynamics, indicating a stronger propagation efficiency of the chain.

The production rates of $H_2O_2$ ($P_{H2O2}$) and $HNO_3$ ($P_{HNO3}$) and their responses to ADE are also summarized in Figure 8 (average for mid-day hours) for the JJJ region (similar illustrations for the other regions can be found in the supplemental Figures S8-S11. Smaller ratios of $P_{H2O2}/P_{HNO3}$ are noted in January compared to July, indicating a stronger VOC-limited regime in January for all regions. The ΔDynamics increases $P_{HNO3}$ but decreases $P_{H2O2}$ in both January and July because the enhanced $NO_x$ at the

surface in a more stable atmosphere likely shifts $O_3$ chemistry towards $NO_x$-rich condition. The ΔPhotolysis reduced both $P_{H2O2}$ and $P_{HNO3}$ but the ratio of $P_{H2O2}/P_{HNO3}$ is decreased due to larger reduction in $P_{H2O2}$ than $P_{HNO3}$. The combined impacts of ΔDynamics and ΔPhotolysis result in a shift towards more VOC-limited conditions in the near-surface layers during both January and July.

**4. Summary**

The impacts of ADE on tropospheric ozone were quantified by using the two-way coupled meteorology and atmospheric chemistry WRF-CMAQ model instrumented with the process analysis methodology. Two manifestations of ADE impacts on $O_3$, changes in atmospheric dynamics (ΔDynamics) and changes in photolysis rates (ΔPhotolysis), were systematically evaluated through simulations that isolated their impacts on modeled process rates over China for winter and summer

conditions (represented by the months of January and July in 2013, respectively). Results suggest that the model performance





for surface $DM1O_3$ simulations improved after the inclusion of ADE which moderately reduced the high-bias in January and low-bias in July. In winter, the inclusion of ADE impacts resulted in an overall reduction in surface $DM1O_3$ across China by up to 39 $\mu g\ m^{-3}$. Changes both in photolysis and atmospheric dynamics due to ADE contributed to the reductions in $DM1O_3$ in winter. In contrast during July, the impact of ADE increased surface $DM1O_3$ across China by up to 4 $\mu g\ m^{-3}$. The

summertime increase in $DM1O_3$ results primarily from ADE induced effects on atmospheric dynamics.  It can thus be postulated that reducing ADE will have potential risk of increasing $O_3$ in winter, but will benefit the reduction of maximum $O_3$ in summer.

Results from IPR analysis suggest that the ADE impacts exhibit strong vertical and diurnal variations. The ADE induced decrease in modeled $DM1O_3$ in January primarily results from ΔPhotolysis which reduced the chemical production of $O_3$ in

the near-ground layers. The increase in $DM1O_3$ in July due to ADE results from a weaker dry deposition sink as well as a stronger chemical source due to higher precursor concentrations in a more stable and shallow PBL. These impacts become stronger under higher $PM_{2.5}$ concentrations when ADE are larger.

The combined ADE impacts reduce $O_x$ in January due to ΔPhotolysis, but slightly increase $O_x$ in July due to ΔDynamics. OH is reduced by ADE in January. However, mid-day OH concentrations during summertime show enhancements associated with

both ΔPhotolysis and ΔDynamics, indicating a stronger mid-day atmospheric oxidizing capacity in July. An increased OH chain length in the near-ground layers is modeled both in January and July, indicating a stronger propagation efficiency of the chain reaction. In both January and July, $P_{HNO3}$ is increased and $P_{H2O2}$ is decreased due to ΔDynamics, and both are reduced due to ΔPhotolysis. The ratio of $P_{H2O2}/P_{HNO3}$ is decreased due to the combined impacts of ΔDynamics and ΔPhotolysis, indicating a shift towards more VOC-limited conditions due to ADE in the near-ground layers during both January and July.

Thus aerosol direct effects on both photolysis rates as well as atmospheric dynamics can impact $O_3$ formation rates and its local and regional distributions. Comparisons of integrated process rates suggest that the decrease in $DM1O_3$ in January results from a larger net chemical sink due to ΔPhotolysis, while the increase in $DM1O_3$ in July is mostly associated with the slower removal due to reduced deposition velocity as well as a stronger photochemistry due to ΔDynamics. The IRR analyses confirm that the process contributions from chemistry to $DM1O_3$ can be influenced by both ΔDynamics and ΔPhotolysis. Reduced

ventilation associated with ΔDynamics enhances the precursor levels, which increase chemical production rate of $O_x$ and OH, resulting in greater $O_3$ chemical formation at noon during both January and July. One the other hand, reduced photolysis rates in ΔPhotolysis results in lower $O_3$ in January. However, in July lower photolysis rates result in accumulation of precursors during the morning hours which eventually lead to higher $O_3$ production at noon.

The comparison of integrated reaction rates from the various simulations also suggest that the increased OH_CL and the shift

towards more VOC-limited conditions are mostly associated with the higher $NO_2$ levels due to ADE. This further emphasizes the importance of $NO_x$ controls in air pollution mitigation. $NO_x$ is a major precursor for both $O_3$ and $PM_{2.5}$. Effective controls



on $NO_x$ will not only gain direct benefits for $O_3$ reduction, but also can indirectly reduce peak $O_3$ through weakening the ADE from the reduced $PM_{2.5}$, highlighting co-benefits from $NO_x$ controls for achieving both $O_3$ and $PM_{2.5}$ reductions.

**Acknowledgements**

This work was supported in part by MEP's Special Funds for Research on Public Welfare (201409002), MOST National Key

R & D program (2016YFC0203306) and MOST National Key R & D program (2016YFC0207601). This work was completed

on the "Explorer 100" cluster system of Tsinghua National Laboratory for Information Science and Technology.

**Disclaimer:** Although this work has been reviewed and approved for publication by the U.S. Environmental Protection Agency, it does not necessarily reflect the views and policies of the agency.

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



**Table 1: Description of sensitivity simulations in this study**

| Short name | Simulation description | Aerosol impacts on photolysis calculations | Aerosol impacts on radiation calculations |
|---|---|---|---|
| SimBL | Baseline simulation | No | No |
| SimNF | No aerosol feedback simulation | Yes | No |
| SimSF | Aerosol feedback simulation | Yes | Yes |





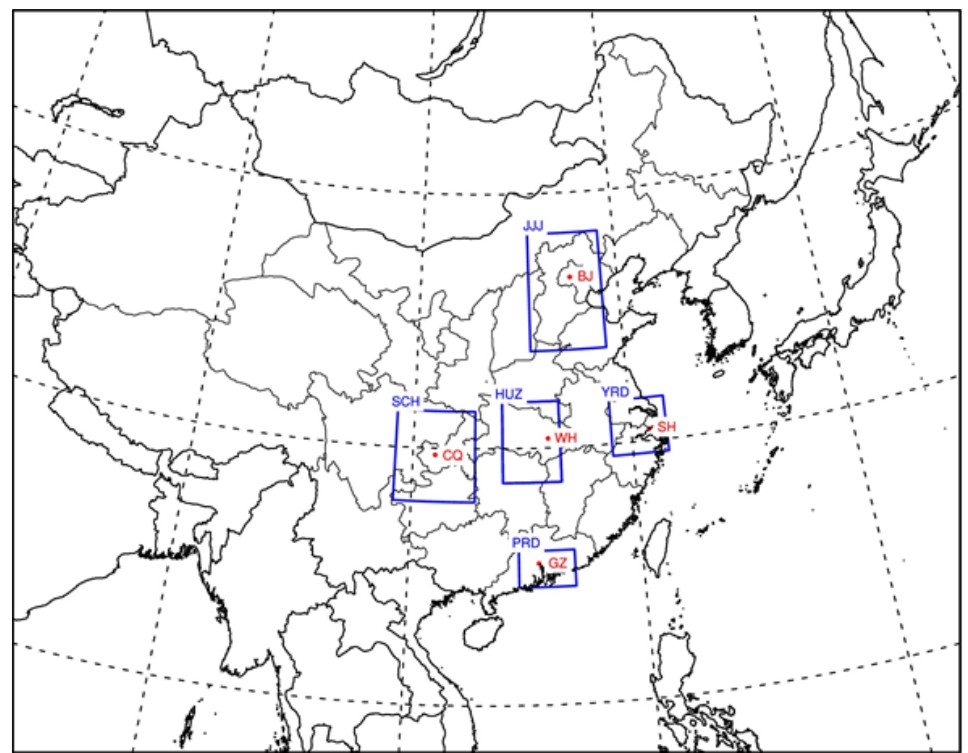

Figure 1: Simulation domain and locations of 5 selected regions in China. Note: JJJ=Jing-Jin-Ji area, YRD=Yangzi-River-Delta area, PRD=Perl-River-Delta area, SCH=Sichuan Basin area, HUZ=Hubei-Hunan area.







**Figure 2: Observed and simulated O₃ and its response to ADE (monthly average of daily 1h maxima, μg m⁻³)**







**Figure 3: Observed and simulated surface O$_3$ concentration against PM$_{2.5}$ concentration (O$_3$ is daily 1h maxima of monitor sites in each region, unit: μg m$^{-3}$; PM$_{2.5}$ is the daily average of those site, unit: μg m$^{-3}$)**





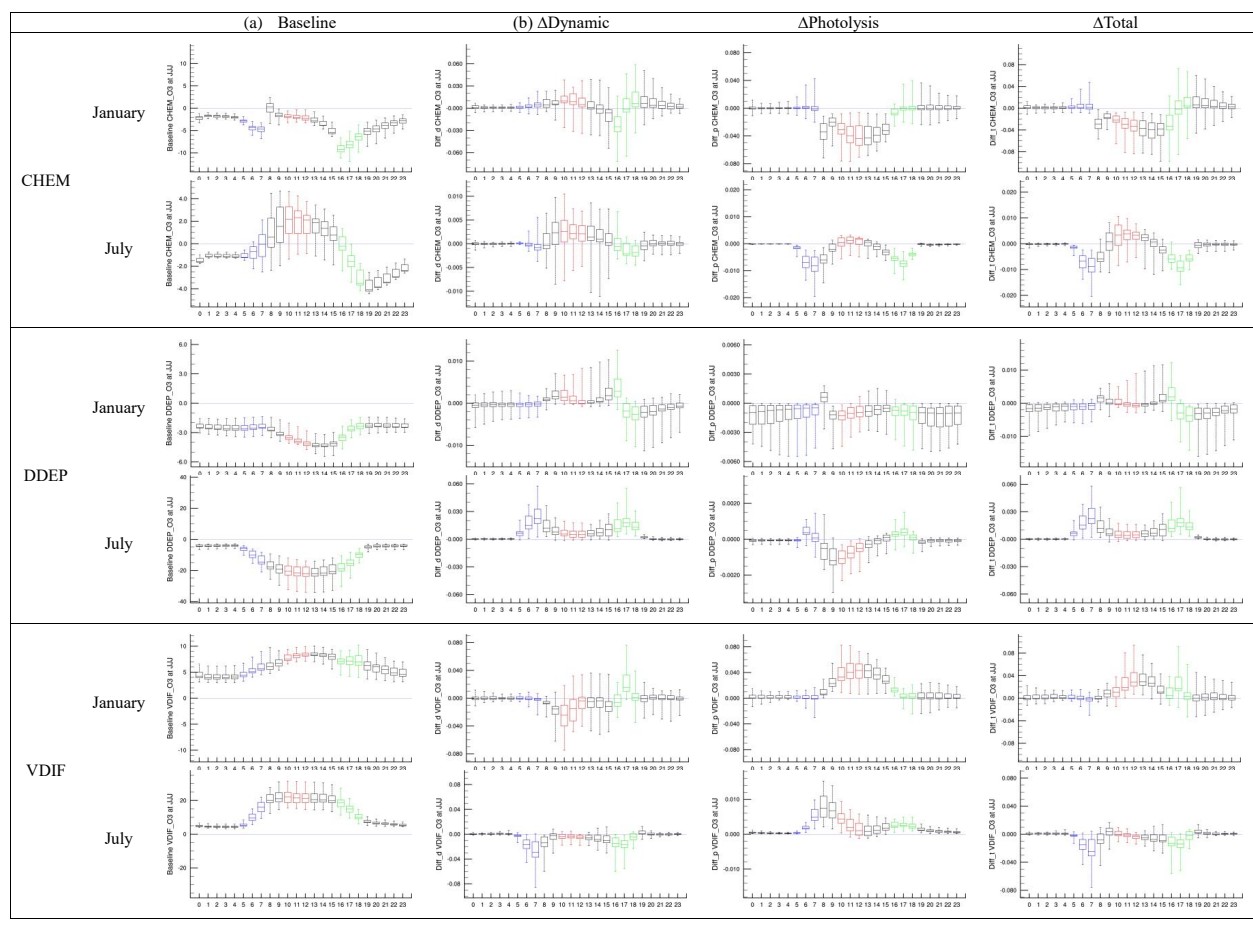



**Figure 4: Diurnal variation of selected integrated process contributions to surface O₃ concentration in JJJ (The calculation is based on the average of grid cells in JJJ; a. Baseline is the simulated O₃ in SimBL, unit: ppb hr⁻¹; b. ΔDynamic is the difference in normalized IPRs between SimSF and SimNF, unit: hr⁻¹; d. ΔPhotolysis is the difference in normalized IPRs between SimNF and SimBL, unit: hr⁻¹; c. ΔTotal is the difference in normalized IPRs between SimSF and SimBL, unit: hr⁻¹, colored bars represent three periods of early morning (blue), noon (red), and late afternoon (green))**







Figure 5: Vertical profile of integrated process contributions to surface O₃ concentration in JJJ (full-layer heights above ground are 40, 96, 160, 241, 355, 503, 688, 884, 1100, 1357m)





Figure 6: Integrated process contributions to daytime near-ground-level O₃ under different PM₂.₅ level in JJJ (between the ground and 350m AGL, model layer 1-5)





**Figure 7: Impacts of ADE on surface O$_x$ and OH (monthly average of noon time 11am-1pm local time)**











**Figure 8: Vertical profile of integrated reaction rates in JJJ at noon (full-layer heights above ground are 40, 96, 160, 241, 355, 503, 688, 884, 1100, 1357m; Baseline is the simulation in SimBL; ΔDynamic is the difference between SimSF and SimNF; ΔPhotolysis is the difference between SimNF and SimBL; ΔTotal is the difference between SimSF and SimBL; $P_{totalOx}$ is total $O_x$ production rate, unit: ppb hr$^{-1}$; OH CL is OH chain length; $P_{NewOH}$ is the production rate of new OH, unit: ppb hr$^{-1}$; $P_{ReactedOH}$ is the production rate of reacted OH, unit: ppb hr$^{-1}$; $P_{H2O2}$ is the production rate of $H_2O_2$, unit: ppb hr$^{-1}$; $P_{HNO3}$ is the production rate of $HNO_3$, unit: ppb hr$^{-1}$; the ratio of $P_{H2O2}/P_{HNO3}$ is only shown for layer 1-5)**