# Peer review of "Impacts of aerosol direct effects on tropospheric ozone through changes in atmospheric dynamics and photolysis rates"

_Atmospheric Chemistry and Physics, 2017_

## Referee Comment (RC1) · Anonymous Referee #1 · 10 Apr 2017

This paper uses WRF-CMAQ simulations to examine the sensitivity of surface ozone in China to aerosol effects on PBL dynamics and photolysis rates, pointing out that aerosol reductions may have inadvertent effects on ozone that need to be considered in air quality management. The simulations appear to have been done carefully and the results are of some moderate interest, but the presentation is very difficult to wade through because of laborious analyses of model results that are of little interest and because of postage-stamp figures that seem like core dumps of obscure information. Also, it appears that the authors did not examine (or mention, unless I missed it) the aerosol effect on ozone through heterogeneous chemistry but I would expect this effect to be at least as large as the effects from dynamics and photolysis. Not accounting for

the effect of aerosols on heterogeneous chemistry (for example through N2O5 hydrolysis) compromises in my opinion the policy-relevant conclusions about the sensitivity of ozone to aerosol reductions. Overall I rate this paper as marginally acceptable in ACP (it would be in my opinion in the bottom third of papers) and then only if the presentation is greatly improved through shortened text and a better selection of figures.

Additional specific comments:

1. Page 2: what meteorological data assimilation is done in the WRF simulation and how would it affect the sensitivity of dynamics to aerosols?

2. Page 4: the authors present as established fact that ADE increases boundary layer stability. I'm not necessarily disputing that but a few references would be helpful.

3. Page 5, line 8: the authors find that aerosols decrease photolysis rates in winter but increase them in summer, and it's not clear why there is this seasonal difference. I suppose indeed that scattering aerosol could increase photolysis rates in summer, but then why not in winter? Line 11 further seems to contradict the statement on line 8 by saying that photolysis rates decrease in summer.

4. Figures 3-6 seem like core dumps. I didn't see a colorbar legend for figures 5-6.

5. Page 6: I don't understand why Δdynamics decreases ozone deposition velocity in summer. During the daytime the ozone deposition velocity should be more limited by the surface resistance.

6. Page 9, line 1: summary states that aerosol effects improve the ozone simulation but I didn't see this demonstrated in the text.

7. Page 9: the summary presents as a punch line that one should decrease NOx emissions to improve both ozone and PM but this is hardly an original result.

---

## Referee Comment (RC2) · Anonymous Referee #2 · 20 Apr 2017

In this paper, the authors have applied the WRF-CMAQ model to analyze the impact of aerosols on tropospheric ozone through their impacts on atmospheric dynamics and photolysis rates. Their results indicate that reducing aerosols may have negative impacts on ozone which need to be considered for air quality management in China. The topic is of general interest given the focus on reducing PM2.5 pollution. The simulations have been designed appropriately to address the goals of the study. However, the authors have not considered the prominent way aerosols impact tropospheric ozone formation - via heterogeneous reactions - which leads me to question the conclusions of this study. Several studies have highlighted the role of aerosols in modulating ozone via heterogeneous reactions (eg., Liao and Seinfeld, 2005; Ti et al., 2005; Pozzoli et al.,

2008; Xu et al., 2012; Lou et al., 2014), which have largely been ignored in this study. The authors need to provide a strong justification for ignoring the impact of aerosols on ozone via heterogeneous reactions, before I can recommend this paper for publication. The presentation of the analysis also needs to be significantly revised, particularly the figures are too small to be legible.

Specific Comments:

P2L10: Please provide a reference for "ADE modulate the temperature, atmospheric ventilation, cloud and rainfall".

P3 Section 2.1: What meteorological fields are used to drive WRF-CMAQ?

P3L5,6: Please define acronyms (e.g., AERO6, BHCOAT) before using them.

P4L20: Please clarify if this is for observations. The observations are hardly visible in Figure 2.

P4L24: Please provide a reference for "In January, O3 production in north China is VOC-limited regime"

P4: It would also be helpful to see maps of PM2.5 to assess if the reductions in O3 due to aerosol feedbacks are co-located with PM concentrations.

P4-5: How significant are the changes in O3 in response to ∆Dynamics and ∆Photolysis

References: H. Liao, J.H. Seinfeld, Global impacts of gas-phase chemistry–aerosol interactions on direct radiative forcing by anthropogenic aerosols and ozone, J. Geophys. Res., 110 (2005), p. D18208 http://dx.doi.org/10.1029/2005JD005907

L. Pozzoli, I. Bey, S. Rast, M.G. Schultz, P. Stier, J. Feichter, Trace gas and aerosol interactions in the fully coupled model of aerosol-chemistry-climate ECHAM5-HAMMOZ: 1. Model description and insights from the spring 2001 TRACE-P experiment, J. Geophys. Res., 113 (2008), p. D07308 http://dx.doi.org/10.1029/2007JD009007

X.X. Tie, S. Madronich, S. Walters, D.P. Edwards, P. Ginoux, N. Mahowald, R.Y. Zhang, C. Lou, G. Brasseur, Assessment of the global impact of aerosols on tropospheric oxidants, J. Geophys. Res, 110 (2005), p. D03204 http://dx.doi.org/10.1029/2004JD005359.

J. Xu, Y.H. Zhang, S.Q. Zheng, Y.J. He, Aerosol effects on ozone concentrations in Beijing: a model sensitivity study, J. Environ. Sci., 24 (4) (2012), pp. 645–656

Sijia Lou, Hong Liao, Bin Zhu, Impacts of aerosols on surface-layer ozone concentrations in China through heterogeneous reactions and changes in photolysis rates, Atmospheric Environment, Volume 85, March 2014, Pages 123-138, ISSN 1352-2310, http://doi.org/10.1016/j.atmosenv.2013.12.004.

---

## Author Comment (AC1) · 23 Jun 2017

We thank the reviewer for the detailed and thoughtful review of our manuscript. Incorporation of the reviewer's suggestion has led to a much improved manuscript. Detailed below is our response to the issues raised by the reviewer. We also detail the specific changes incorporated in the revised manuscript in response to the reviewer's comments.

[Comment]: The simulations appear to have been done carefully and the results are of some moderate interest, but the presentation is very difficult to wade through because of laborious analyses of model results that are of little interest and because of postagestamp figures that seem like core dumps of obscure information.

[Response]: We thank the reviewer for the suggestion. To improve the flow of the manuscript, we have reduced the amount of information presented and also moved some of the information into the supplementary material. Furthermore, we have increased the size of several figures to better present the information in the revised manuscript.

[Comment]: it appears that the authors did not examine (or mention, unless I missed it) the aerosol effect on ozone through heterogeneous chemistry but I would expect this effect to be at least as large as the effects from dynamics and photolysis. Not accounting for the effect of aerosols on heterogeneous chemistry (for example through N2O5 hydrolysis) compromises in my opinion the policy-relevant conclusions about the sensitivity of ozone to aerosol reductions.

[Response]: We agree with the reviewer that the heterogeneous reactions associated with aerosols have substantial impacts on ozone, including hydrolysis of N2O5, irreversible absorption of NO2 and NO3 as well as the uptake of HO2, and are well documented in the literature (Tang et al., 2004; Tie et al., 2005; Liao and Seinfeld, 2005; Pozzoli et al., 2008; Li et al., 2011; Xu et al., 2012; Lou et al., 2014). Our model contains comprehensive treatment of heterogeneous hydrolysis of N2O5 (Davis et al., 2008; Sarwar et al., 2012; Sarwar et al., 2014). While our model accounts for such a heterogeneous reaction, we have not quantified its impacts on ozone in this study.

However, in this study, we focused our analysis on another important aspect of aerosol influence (ADE, the aerosol direct effects), i.e., scattering and absorption of incoming solar radiation and how the subsequent effects of the associated cooling suppresses atmospheric ventilation. The assessment of impacts of aerosol direct effects (ADE) is an important aspect of designing emission reduction strategies that seek co-benefits associated with reductions in both particulate matter and ozone. In this study, we examine the ADE impacts which were not well quantified in previous studies. It may
be noted that all model calculations analyzed in this study included the heterogeneous N2O5 hydrolysis pathway.

We agree with the reviewer that all influences of aerosols on ozone need to be addressed before definite policy-relevant conclusions regarding their overall impact can be reached. China plans to implement stringent control actions aimed at lowering the ambient concentrations of PM2.5 in the next two decades. It will be necessary to quantify all the possible influences resulting from this expected reduction of aerosols, including changes in heterogeneous reactions associated with aerosols, as well as the changes expected in ADE discussed in this manuscript. In addition, secondary aerosol and ozone comes from both NOx and VOCs emissions. Reducing aerosols by controlling gaseous precursors will also have substantial impacts on ozone (Xing et al., 2017), which needs further evaluation.

To address the reviewer's concern, we have clarified the scope of our analysis in the revised manuscript as below:

(Page 2 Line 21-25) "Many studies suggest that aerosols may have substantial impacts on ozone through heterogeneous reactions including hydrolysis of N2O5, irreversible absorption of NO2 and NO3, as well as the uptake of HO2 (Tang et al., 2004; Tie et al., 2005; Li et al., 2011; Lou et al., 2014). While our model contains comprehensive treatment of the heterogeneous hydrolysis of N2O5 (Davis et al., 2008; Sarwar et al., 2012; Sarwar et al., 2014), we have not quantified its impacts on ozone in this study. However, ADE impacts on ozone have not been well evaluated previously."

(Page 10 Line 15-19) "Reducing aerosols will have substantial impacts on ozone. Quantification of the aerosol influence on ozone is important to understand co-benefits associated with reductions in both particulate matter and ozone. This study focused on the evaluation of ADE impacts which were not well quantified previously. However, the heterogeneous reactions associated with aerosols, as well as the impacts of emission controls of gaseous precursors on both aerosols and ozone also need to be studied in

order to fully understand the influence from reducing aerosols on ambient ozone."

Reference

Li J., Wang Z., Wang X., Yamaji K., et al. Impacts of aerosols on summertime tropospheric photolysis frequencies and photochemistry over Central Eastern China. Atmospheric Environment, 2011, 45: 1817-1829.

Liao, H., Seinfeld, J.H., Global impacts of gas-phase chemistry–aerosol interactions on direct radiative forcing by anthropogenic aerosols and ozone, J. Geophys. Res., 2005, 110, D18208

Tang Y. H., Carmichael G. R., Kurata G., Uno I., et al. Impacts of dust on regional tropospheric chemistry during the ACE-Asia experiment: A model study with observations. Journal of Geophysical Research-Atmospheres, 2004, 109

Tie X. X., Madronich S., Walters S., et al. Assessment of the global impact of aerosols on tropospheric oxidants. Journal of Geophysical Research-Atmospheres 2005, 110

Lou S. J., Liao H., Zhu B. Impacts of aerosols on surface-layer ozone concentrations in China through heterogeneous reactions and changes in photolysis rates. Atmospheric Environment, 2014, 85: 123-138.

Pozzoli, L., Bey, I., Rast, S., Schultz, M.G., Stier, P., Feichter, J., Trace gas and aerosol interactions in the fully coupled model of aerosol-chemistry-climate ECHAM5-HAMMOZ: 1. Model description and insights from the spring 2001 TRACE-P experiment, J. Geophys. Res., 2008, 113, D07308

Xu, J., Zhang, Y.H., Zheng, S.Q., He, Y.J., Aerosol effects on ozone concentrations in Beijing: a model sensitivity study, J. Environ. Sci., 2012, 24 (4), 645–656

Davis, J. M., Bhave, P. V., and Foley, K. M.: Parameterization of N2O5 reaction probabilities on the surface of particles containing ammonium, sulfate, and nitrate, Atmos. Chem. Phys., 8, 5295–5311, doi:10.5194/acp-8-5295-2008, 2008.

Sarwar, G., Simon, H., Bhave, P., and Yarwood, G.: Examining the impact of heterogeneous nitryl chloride production on air quality across the United States, Atmospheric Chemistry & Physics, 12, 1-19, 2012.

Sarwar, G., Simon, H., Xing, J., Mathur, R.: Importance of tropospheric ClNO2 chemistry across the Northern Hemisphere, Geophysical Research Letters, 41, 4050-4058, 2014.

Xing, J., Wang, S. X., Jang, C., et al. Overview of ABaCAS: an air pollution control cost-benefit and attainment assessment system and its applications in China, EM of Air & Waste Management Association, 2017 April.

[Comment]: Page 2: what meteorological data assimilation is done in the WRF simulation and how would it affect the sensitivity of dynamics to aerosols?

[Response]: We followed our previous coupled model design (Xing et al., 2015). The strength of nudging coefficients for four-dimensional data assimilation and indirect soil temperature nudging employed in WRF have been tested and chosen to improve model performance for meteorological variables without dampening the effects of radiative feedbacks. The nudging coefficient for both u/v-wind and potential temperature is set to 0.00005 s-1, while 0.00001 s-1 is used for nudging of the water vapor mixing ratio.

We have clarified it in the revised manuscript as below:

(Page 3 Line 7-12) "The meteorological inputs for WRF simulations were derived from the NCEP FNL (Final) Operational Global Analysis data which has 1 degree spatial and 6-hour temporal resolution. NCEP ADP Operational Global Surface Observations were used for surface reanalysis and four dimensional data assimilation. We have tested and chosen proper strength of nudging coefficients, i.e., 0.00005 sec-1 is used for nudging of both u/v-wind and potential temperature, 0.00001 sec-1 is used for nudging of water vapor mixing ratio, to improve model performance without dampening the effects of radiative feedbacks (Hogrefe et al., 2015; Xing et al., 2015)."

[Figure]

Reference

Hogrefe, C., Pouliot, G., Wong, D., Torian, A., Roselle, S., Pleim, J. and Mathur, R.: Annual application and evaluation of the online coupled WRF–CMAQ system over North America under AQMEII phase 2. Atmospheric Environment, 115, 683-694, 2015.

Xing, J., Mathur, R., Pleim, J., Hogrefe, C., Gan, C.M., Wong, D.C. and Wei, C.: Can a coupled meteorology–chemistry model reproduce the historical trend in aerosol direct radiative effects over the Northern Hemisphere?. Atmospheric Chemistry and Physics, 15(17), 9997-10018, 2015.

[Comment]: Page 4: the authors present as established fact that ADE increases boundary layer stability. I'm not necessarily disputing that but a few references would be helpful.

[Response]: Both aerosol scattering and absorption of incoming solar radiation result in reduced solar radiation impinging the ground causing reduced ground temperatures, while light-absorbing carbon aloft increases the temperature in the upper boundary layer, but cools the surface. This cooling increases stability of the boundary layer and reduces ventilation of pollutants in the boundary layer.

Following the reviewer's suggestion, in the revised manuscript we have included a few references analyzing this process chain. The manuscript discussion has been updated as follows:

(Page 5 Line 6-8) "In January, $O_3$ production in north China is occurring in a VOC-limited regime, thus increases in NOx at the surface stemming from the stabilized atmosphere by ADE (Jacobson et al., 2007; Mathur et al., 2010; Ding et al., 2013; Xing et al., 2015) inhibit $O_3$ formation due to enhanced titration by NO."

Reference

Jacobson, M. Z.; Kaufman, Y. J.; Rudich, Y. Examining feedbacks of aerosols to urban climate with a model that treats 3-D clouds with aerosol inclusions. J. Geophys. Res.

2007, 112, D24205

Mathur, R.; Pleim, J. E.; Wong, D. C.; Otte, T. L.; Gilliam, R. C.; Roselle, S. J.; Young, J. O.; Binkowski, F. S.; Xiu, A. The WRF-CMAQ Integrated On-Line Modeling System: Development, Testing, and Initial Applications. In Air Pollution Modeling and its Applications XX; Steyn, D. G.; Rao, S. T., Eds.; Springer: Netherlands, Netherlands, 2010; pp 155−159.

Ding AJ, Fu CB, Yang XQ, Sun JN, Petaja T, Kerminen VM, et al. Intense atmospheric pollution modifies weather: a case of mixed biomass burning with fossil fuel combustion pollution in eastern China. Atmospheric Chemistry and Physics 2013; 13: 10545-10554.

Xing, J.; Mathur, R.; Pleim, J.; Hogrefe, C.; Gan, C. M.; Wong, D. C.; Wei, C.; Wang, J. Air pollution and climate response to aerosol direct radiative effects: a modeling study of decadal trends across the northern hemisphere. J. Geophys. Res. 2015, 120, 12,221−12,236.

[Comment]: Page 5, line 8: the authors find that aerosols decrease photolysis rates in winter but increase them in summer, and it's not clear why there is this seasonal difference. I suppose indeed that scattering aerosol could increase photolysis rates in summer, but then why not in winter?

[Response]: We agree with the reviewer that the scattering aerosol could increase photolysis rates in both seasons, while absorbing aerosol act oppositely. The overall impacts on photolysis rates are depend on the combined effects of all types of aerosols. Similar results were found in Tie et al (2005) who reported that surface-layer photolysis rates in eastern China were reduced less significantly in summer than in winter.

In this study, we found that the response of photolysis rates to ADE presents a strong diurnal pattern, which shows reductions in the early morning and late afternoon, but shows a slight increase at noon. The reason might be associated with the enhanced

ambient precursor concentrations (due to less reaction in early morning) at noon when O3 reaches the daily maximum.

To address the reviewer's concern, we have clarified this discussion in the revised manuscript as below:

(Page 5 Line 20-21) "Similar results were found in Tie et al (2005) who reported that surface-layer photolysis rates in eastern China were reduced less significantly in summer than in winter."

(Page 5 Line 24-26) "This increase in precursor concentrations then leads to enhanced O3 formation later in the day which compensates for or even overwhelms the disbenefit from the reduced solar radiances."

Reference

Tie X. X., Madronich S., Walters S., et al. Assessment of the global impact of aerosols on tropospheric oxidants. Journal of Geophysical Research-Atmospheres 2005, 110

[Comment]: Line 11 further seems to contradict the statement on line 8 by saying that photolysis rates decrease in summer

[Response]: The photolysis reaction rate depends on solar radiances and precursor levels. We agree with the reviewer that our original statement is ambiguous. We clarified it as below:

(Page 5 Line 24-26) "This increase in precursor concentrations then leads to enhanced O3 formation later in the day which compensates for or even overwhelms the disbenefit from the reduced solar radiances."

[Comment]: Figures 3-6 seem like core dumps. I didn't see a colorbar legend for figures 5-6.

[Response]: We have reduced the content of these figures to focus on the most important aspects and added a colorbar legend for figures 5-6 in the revised manuscript.

[Comment]: Page 6: I don't understand why dynamics decreases ozone deposition velocity in summer. During the daytime the ozone deposition velocity should be more limited by the surface resistance.

[Response]: The dry deposition velocity is computed as the reciprocal of the sum for aerodynamic resistance (Ra), quasi-laminar resistance (Rb), and surface resistance (Rc). The Ra is a function of wind speed and turbulence. Since the changes in dynamics decrease the wind speed, they thus increase Ra and consequently reduce dry deposition."

We have clarified this in the revised manuscript as below:

(Page 6 Line 23-25) "The stabilization of the atmosphere due to Dynamics leads to lower dry deposition rates (due to lower dry deposition velocity from the enhanced aerodynamic resistance) and thus increases surface O3."

[Comment]: Page 6: Page 9, line 1: summary states that aerosol effects improve the ozone simulation but I didn't see this demonstrated in the text.

[Response]: Following the reviewer's suggestion, we have reduced the content of Figure 3 and summarized the comparison in Table 2 in the revised manuscript, as below:

(Page 5 Line 28- Page 6 Line 10) "The impact of the ADE on O3 is further explored by examining the relationship between the observed and simulated O3 concentrations (DM1O3, daily values of the cities located in China) as a function of the observed PM2.5 concentrations (observed daily averaged values in those cities), as displayed in Figure 3. The predicted ozone concentrations under both low- and high- PM2.5 levels are compared in Table 2. In regards to model performance for DM1O3 simulations, the model generally exhibits a slight high bias in January but a low bias in July across the 5 regions. The inclusion of ADE moderately reduced O3 concentrations in January and slightly increased O3 in July, resulting in reduction in bias and improved performance for DM1O3 simulation in both January and July for most of regions.

Interestingly, from low to moderate PM2.5 levels (i.e., PM2.5 < 120 ug m-3), higher O3 concentration occur with higher PM2.5 concentrations, which is evident in both observations and simulations, suggestive of common precursors (e.g., NOx), source sectors, and/or transport pathways contributing to both O3 and PM2.5 in these regions. However, a negative correlation between O3 and PM2.5 is evident in winter when the PM2.5 can reach high levels larger than 120 ug m-3, indicating the strong ADE impacts on O3 through both feedbacks to dynamics and photolysis which significantly reduced O3."

[Comment]: Page 9: the summary presents as a punch line that one should decrease NOx emissions to improve both ozone and PM but this is hardly an original result.

[Response]: Traditionally, the co-benefits from NOx controls for ozone and PM reductions were thought of resulting from the fact that NOx acts as a common precursor for both O3 and PM2.5, thus a decrease of NOx emissions would be expected to reduce both ozone and PM. The analyses presented in this study reveals that the consideration of NOx controls is not only beneficial for directly reducing PM2.5 and O3, but also because of indirect benefits in reducing peak O3 through the weakening of the ADE from the reduced PM2.5, increasing the co-benefits from NOx controls for achieving both O3 and PM2.5 reductions.

To address the reviewer's concern, we have clarified this in the revised manuscript as below:

(Page 10 Line 11-14) "Traditionally, the co-benefits from NOx control for ozone and PM reduction are mostly thought of as resulting from the fact that NOx is a common precursor for both O3 and PM2.5. This study suggests that effective controls on NOx will not only gain direct benefits for O3 reduction, but also can indirectly reduce peak O3 through weakening the ADE from the reduced PM2.5, increasing the estimated co-benefits from NOx controls for achieving both O3 and PM2.5 reductions."

Please also note the supplement to this comment:
http://www.atmos-chem-phys-discuss.net/acp-2017-198/acp-2017-198-AC1-supplement.pdf

---

## Author Comment (AC2) · 23 Jun 2017

We thank the referee for a thoughtful and detailed review of our manuscript. Incorporation of the reviewer's suggestions has led to a much improved manuscript. Below we provide a point-by-point response to the reviewer's comments and summarize the changes that have been incorporated in the revised manuscript.

[Comment]: In this paper, the authors have applied the WRF-CMAQ model to analyze the impact of aerosols on tropospheric ozone through their impacts on atmospheric dynamics and photolysis rates. Their results indicate that reducing aerosols may have negative impacts on ozone which need to be considered for air quality management

in China. The topic is of general interest given the focus on reducing PM2.5 pollution. The simulations have been designed appropriately to address the goals of the study.

[Response]: We thank the reviewer for the overall positive assessment of the manuscript and recognition of the implications of the results of the analysis presented.

[Comment]: However, the authors have not considered the prominent way aerosols impact tropospheric ozone formation - via heterogeneous reactions - which leads me to question the conclusions of this study. Several studies have highlighted the role of aerosols in modulating ozone via heterogeneous reactions (eg., Liao and Seinfeld, 2005; Ti et al., 2005; Pozzoli et al., 2008; Xu et al., 2012; Lou et al., 2014), which have largely been ignored in this study. The authors need to provide a strong justification for ignoring the impact of aerosols on ozone via heterogeneous reactions, before I can recommend this paper for publication.

[Response]: We agree with the reviewer that the heterogeneous reactions associated with aerosols have substantial impacts on ozone, including hydrolysis of N2O5, irreversible absorption of NO2 and NO3 as well as the uptake of HO2, and are well documented in the literature (Tang et al., 2004; Tie et al., 2005; Liao and Seinfeld, 2005; Pozzoli et al., 2008; Li et al., 2011; Xu et al., 2012; Lou et al., 2014). Our model contains comprehensive treatment of heterogeneous hydrolysis of N2O5 (Davis et al., 2008; Sarwar et al., 2012; Sarwar et al., 2014). While our model accounts for such a heterogeneous reaction, we have not quantified its impacts on ozone in this study.

However, in this study, we focused our analysis on another important aspect of aerosol influence (ADE, the aerosol direct effects), i.e., scattering and absorption of incoming solar radiation and how the subsequent effects of the associated cooling suppresses atmospheric ventilation. The assessment of impacts of aerosol direct effects (ADE) is an important aspect of designing emission reduction strategies that seek co-benefits associated with reductions in both particulate matter and ozone. In this study, we examine the ADE impacts which were not well quantified in previous studies. It may

be noted that all model calculations analyzed in this study included the heterogeneous N2O5 hydrolysis pathway. We agree with the reviewer that all influences of aerosols on ozone need to be addressed before definite policy-relevant conclusions regarding their overall impact can be reached. China plans to implement stringent control actions aimed at lowering the ambient concentrations of PM2.5 in the next two decades. It will be necessary to quantify all the possible influences resulting from this expected reduction of aerosols, including changes in heterogeneous reactions associated with aerosols, as well as the changes expected in ADE discussed in this manuscript. In addition, secondary aerosol and ozone comes from both NOx and VOCs emissions. Reducing aerosols by controlling gaseous precursors will also have substantial impacts on ozone (Xing et al., 2017), which needs further evaluation.

To address the reviewer's concern, we have clarified the scope of our analysis in the revised manuscript as below:

(Page 2 Line 21-25) "Many studies suggest that aerosols may have substantial impacts on ozone through heterogeneous reactions including hydrolysis of N2O5, irreversible absorption of NO2 and NO3, as well as the uptake of HO2 (Tang et al., 2004; Tie et al., 2005; Li et al., 2011; Lou et al., 2014). While our model contains comprehensive treatment of the heterogeneous hydrolysis of N2O5 (Davis et al., 2008; Sarwar et al., 2012; Sarwar et al., 2014), we have not quantified its impacts on ozone in this study. However, ADE impacts on ozone have not been well evaluated previously."

(Page 10 Line 15-19) "Reducing aerosols will have substantial impacts on ozone. Quantification of the aerosol influence on ozone is important to understand co-benefits associated with reductions in both particulate matter and ozone. This study focused on the evaluation of ADE impacts which were not well quantified previously. However, the heterogeneous reactions associated with aerosols, as well as the impacts of emission controls of gaseous precursors on both aerosols and ozone also need to be studied in order to fully understand the influence from reducing aerosols on ambient ozone."

[Figure]

[Comment]: P4: It would also be helpful to see maps of PM2.5 to assess if the reductions in O3 due to aerosol feedbacks are co-located with PM concentrations.

[Response]: Following the reviewer's suggestion, we made the plot of PM2.5 concentrations as Figure R1 It clearly shows that the ADE effects on O3 are co-located with PM concentrations. Besides, we calculated the O3 responses to ADE under low- and high-PM2.5 levels, summarized in Table 2. Mostly, the O3 responses to ADE are larger under high PM2.5 levels, indicating the positive correlations between O3 responses and PM levels.

We have added corresponding discussions to the revised manuscript, as below:

(Page 6 Line 3-5) "Comparing the O3 responses to ADE (see $\Delta$-ADE in Table 2) under low- and high- PM2.5 levels, reveals that O3 responses to ADE are larger under high PM2.5 levels, indicating the positive correlations between O3 responses and PM2.5 levels."

[Comment]: P4-5: How significant are the changes in O3 in response to Dynamics and Photolysis

[Response]: Out results suggest that ADE reduced surface daily maxima 1h O3 (DM1O3) in China by up to 39 $\mu$g m-3 through the combination of $\Delta$Dynamics and $\Delta$Photolysis in January, but enhanced surface DM1O3 by up to 4 $\mu$g m-3 in July.

The enhancement in peak O3 in summer found in this study is different from the traditional expectation of a negative O3 response to ADE when solely considering the reduction in solar radiance.

Please also note the supplement to this comment:
http://www.atmos-chem-phys-discuss.net/acp-2017-198/acp-2017-198-AC2-supplement.pdf
* * *
[Figure]

Figure R1 spatial distribution of PM$_{2.5}$ concentration and O$_3$ response to ADE

**Fig. 1.**

**Supplement:**

**Impacts of aerosol direct effects on tropospheric ozone through changes in atmospheric dynamics and photolysis rates**

Jia Xing1, Jiandong Wang1, Rohit Mathur2, Shuxiao Wang1, Golam Sarwar2, Jonathan Pleim2, Christian

5 Hogrefe2, Yuqiang Zhang2, Jingkun Jiang1, David C. Wong2, Jiming Hao1

[revised manuscript text omitted]

The impact of the ADE on  $O_3$  is further explored by examining the relationship between the observed and simulated  $O_3$  concentrations (DM1O3, daily values of the cities located in China) as a function of the observed PM2.5 concentrations (observed daily averaged values in those cities), as displayed in Figure 3. The predicted ozone concentrations under both lowand high- PM2.5 levels are compared in Table 2. In regards to model performance for DM1O3 simulations, the model generally

5

exhibits a slight high bias in January but a low bias in July across the 5 regions. The inclusion of ADE moderately reduced  $O_3$  concentrations in January and slightly increased  $O_3$  in July, resulting in reduction in bias and improved performance for DM1O3 simulation in both January and July for most of regions. Comparing the  $O_3$  responses to ADE (see  $\Delta$ -ADE in Table 2) under low- and high- PM2.5 levels, reveals that  $O_3$  responses to ADE are larger under high PM2.5 levels, indicating the positive correlations between  $O_3$  responses and PM2.5 levels.

5 co

10

Interestingly, from low to moderate  $PM_{2.5}$  levels (i.e.,  $PM_{2.5} < 120 \ \mu g \ m^{-3}$ ), higher O3 concentration occur with higher  $PM_{2.5}$  concentrations, which is evident in both observations and simulations, suggestive of common precursors (e.g., NOx), source sectors, and/or transport pathways contributing to both O3 and PM2.5 in these regions. However, a negative correlation between O3 and PM2.5 is evident in winter when the PM2.5 can reach high levels larger than 120  $\mu g \ m^{-3}$ , indicating the strong ADE impacts on O3 through both feedbacks to dynamics and photolysis which significantly reduced O3.

**3.2 IPRs response to ADE**

To further explore the ADE impacts on simulated  $O_3$ , the integrated process contributions are further analyzed in three ways: (a) 24-hour diurnal variations of process contributions to simulated surface  $O_3$  (Figure 4), (b) vertical profiles from ground up

to 1357 m AGL (above ground level, in model layer 1-10) at noon (Figure 5), and (c) correlations with near-ground  $PM_{2.5}$ (average concentrations between the ground and 355m AGL, model layer 1-5) (Figure 6). In the following, we limit our discussion to analysis of model results for the JJJ region which exhibited the strongest ADE among the regions; similar results were found for the other 4 regions and can be found in the Supporting Information section.

Diurnal variation of process contributions from chemistry (CHEM), dry deposition (DDEP) and vertical turbulent mixing
 (VDIF) which together contribute to more than 90% of the O3 rate of change for the JJJ region, are illustrated in Figure 4. The diurnal variation of IPRs for other processes and their response to ADE are displayed in Figure S1 for JJJ and Figure S2-S5 for other 4 regions.

For surface-level  $O_3$ , VDIF is the major source and DDEP is the major sink (Figure S1). The stabilization of atmosphere due to  $\Delta D$ ynamics leads to lower dry deposition rates (due to lower dry deposition velocity from the enhanced aerodynamic

- resistance) and thus increases surface O3. The largest impact of ΔDynamics on DDEP occurs during early morning and late afternoon which is consistent with the response of the PBL height to ADE noted in our previous analysis (Xing et al., 2015). Expectedly, CHEM is the second largest sink for surface O3 during January, but a source for surface O3 during the daytime in July. The ΔDynamics increased the surface O3 around noon in both January and July for almost all regions (no impacts in PRD and YRD in January, see Figure S2-S3), since increased stability due to ΔDynamics concentrated more precursors locally,
  leading to enhanced O3 formation during the photochemically most active period of the day. The ΔDynamics reduced the
- surface O3 around late afternoon in January at all regions. This is because the increased atmospheric stability during late

afternoon and evening hours increased NOx concentration which titrated more O3. The  $\Delta$ Photolysis reduced surface O3 in all regions in January. These reductions were more pronounced during the early morning hours when the photolysis rate are most sensitive to the radiation intensity. The  $\Delta$ Photolysis resulted in comparatively larger reductions in surface O3 during the early morning and late afternoon hours in July, but slightly increased surface O3 at noon for most of the regions. This increase in O3

5

can be hypothesized to result from the following sequence of events. Slower photochemical reaction in the morning in the  $\Delta$ Photolysis case lead to higher levels of precursors, whose accumulation then enhances O3 formation at noon. This hypothesis is further confirmed by the changes in the diurnal variation of NO2 which suggest that higher NO to NO2 conversion during early morning results in enhanced daytime NO2 levels (see Figure S6), consequently leading to higher noon-time O3.

For aloft  $O_3$  (from 100 to 1600 meters above ground) as seen in Figure 5, CHEM is the major source for  $O_3$  at noon both in

- 10 January and in July. The ΔDynamics increased near-surface O3 (below 500m, model layer 1-6), but reduced upper-level O3 (above 500m, model layer 7-10), because increased stability of the atmosphere concentrated precursors emissions within a shallower layer resulting in higher O3 production. The ΔPhotolysis case considerably reduced near-surface O3 at noon in January. In July, ΔPhotolysis increased upper-level O3 at noon. Higher levels of precursors at noon might be the reason for such enhancement (see Figure S6).
- 15 The daytime near-ground-averaged (between the ground and 350m AGL, layers 1-5) IPR responses to ADE are shown in Figure 6 for JJJ and in Figure S7 for other regions. The IPR and its responses are presented as a function of near-ground-averaged PM2.5 concentrations. As shown in Figure 6, as PM2.5 concentrations increase, the positive contribution of CHEM in July become larger while the negative contribution of CHEM in January become smaller. The overall ADE enhanced CHEM and thus increased O3 concentration in July, and such enhancement are generally larger for higher PM2.5 loading. In contrast,
- 20 in January overall ADE resulted in higher rates of  $O_3$  destruction due to chemistry (negative contribution of CHEM), and the magnitude of this sink increased as  $PM_{2.5}$  concentrations increase. The reduction of  $O_3$  stemming from the enhancements in the chemical sinks is the dominant impact of ADE in January. The enhanced positive contribution of CHEM due to  $\Delta D$ ynamics was partially compensated by the reduction from  $\Delta P$ hotolysis (see Figure S7), resulting in a slight increase in the positive CHEM contribution to  $O_3$  in July.
- 25 DDEP is the major sink of daytime O3 during both January and July. The increased stability due to ADE reduced deposition velocity and thus increases O3. These effects become larger with increasing PM2.5 concentrations. Thus, weaker removal of O3 from DDEP associated with ADE, contributed to higher O3 in most regions during both January and July. Enhanced O3 source of CHEM and reduced O3 sink of DDEP is the dominant impact of ADE in July.

**3.3 IRR response to ADE**

30 The simulated mid-day average (11:00-13:00 local time) surface  $O_x$  (defined as the sum of O, O3, NO2, NO3, N2O5, HNO3, PNA, NTR, PAN and PANX) and OH and their responses to ADE are shown in Figure 7. Both Ox and OH are significantly

reduced in the  $\Delta$ Photolysis case in January throughout the modeling domain. Both Ox and OH also show reductions in the middle portions of east China in the  $\Delta$ Dynamics case in January. Together, the combined ADE impacts result in reduced Ox and OH in January, with widespread reductions primarily due to ADE on photolysis. In July,  $\Delta$ Photolysis increased mid-day OH across most of China (Figure 7) which is consistent with the increase of O3 at noon stemming from a higher level of precursors accumulation due to  $\Delta$ Photolysis. The overall ADE impact on OH is controlled by  $\Delta$ Photolysis, and result in

increased mid-day OH across most of China. For  $O_x$ , however, the impact of  $\Delta Dynamics$  overwhelms the impact from

5

10

 $\Delta$ Photolysis, resulting in increase in Ox concentrations in east China including YRD, SCH and HUZ.

To further examine the response of  $O_x$  to ADE, in Figure 8 we examine vertical profiles of the integrated reaction rates at noon for the JJJ region. The stabilization of the atmosphere due to  $\Delta$ Dynamics concentrates precursors within a lower PBL, resulting in an increased total  $O_x$  production rate ( $P_{totalOx}$ ) mostly in near-ground model layers (below 500m, model layer 1-6); in magnitude aloft (above 500m, model layer 7-10), this change in  $P_{totalOx}$  is smaller in January, and become decreasing in July. The reduction of  $P_{totalOx}$  due to  $\Delta$ Photolysis is greatest at the surface in January, and declines with altitude, and even becomes reversed at high layers (about 1300m, model layer 10) (Figure 8a). The overall ADE impact in January is mainly dominated by  $\Delta$ Photolysis which largely overwhelms the impact of  $\Delta$ Dynamics (Figure 8a). However, in July,  $\Delta$ Photolysis enhanced

[revised manuscript text omitted]

5 that the process contributions from chemistry to  $DM1O_3$  can be influenced by both  $\Delta D$ ynamics and  $\Delta P$ hotolysis. Reduced ventilation associated with  $\Delta$ Dynamics enhances the precursor levels, which increase chemical production rate of Ox and OH, resulting in greater O3 chemical formation at noon during both January and July. One the other hand, reduced photolysis rates in  $\Delta$ Photolysis results in lower O3 in January. However, in July lower photolysis rates result in accumulation of precursors during the morning hours which eventually lead to higher O3 production at noon.

- 10 The comparison of integrated reaction rates from the various simulations also suggest that the increased OH CL and the shift towards more VOC-limited conditions are mostly associated with the higher NO2 levels due to ADE. This further emphasizes the importance of NOx controls in air pollution mitigation. Traditionally, the co-benefits from NOx control for ozone and PM reduction are mostly because that NOx is a common precursor for both O3 and PM2.5. This study suggests that effective controls on NOx will not only gain direct benefits for O3 reduction, but also can indirectly reduce peak O3 
[revised manuscript text omitted]

|         |       | Low PM 2.5 (<60µg m -3 ) |                      |       |                | High PM 2.5 (>60µg m -3 ) |                       |       |       |               |                       |
|---------|-------|------------------------------------------------|----------------------|-------|----------------|-------------------------------------------------|-----------------------|-------|-------|---------------|-----------------------|
| Region  |       | OBS                                            | Normalized Mean Bias |       | $\Delta$ -ADE* | OBS                                             | Normalized Mean Bias  |       |       | $\Delta$ -ADE |                       |
|         |       | (µg m -3 )                          | SimSF                | SimNF | SimBL          | (µg m -3 )                           | (µg m -3 ) | SimSF | SimNF | SimBL         | (µg m -3 ) |
| January | JJJ   | 62.52                                          | 3%                   | 4%    | 5%             | -1.05                                           | 37.02                 | 22%   | 36%   | 53%           | -11.36                |
|         | YRD   | 63.89                                          | 38%                  | 41%   | 43%            | -2.76                                           | 66.74                 | 54%   | 59%   | 67%           | -8.85                 |
|         | PRD   | 97.25                                          | 25%                  | 26%   | 29%            | -4.52                                           | 122.61                | 6%    | 5%    | 9%            | -4.63                 |
|         | HUZ   | 47.67                                          | 172%                 | 173%  | 193%           | -10.17                                          | 67.29                 | 107%  | 125%  | 142%          | -23.9                 |
|         | SCH   | 88.63                                          | -43%                 | -40%  | -38%           | -3.85                                           | 111.19                | -5%   | 2%    | 8%            | -13.78                |
|         | China | 76.61                                          | 30%                  | 31%   | 34%            | -2.96                                           | 62.68                 | 42%   | 48%   | 56%           | -8.61                 |
| July    | JJJ   | 159.27                                         | -29%                 | -28%  | -28%           | -0.51                                           | 178.54                | -25%  | -25%  | -25%          | 1.02                  |
|         | YRD   | 171.04                                         | -31%                 | -31%  | -32%           | 0.84                                            | 233.13                | -24%  | -25%  | -23%          | -0.51                 |
|         | PRD   | 129.02                                         | -20%                 | -19%  | -20%           | -0.09                                           | 312.21                | -44%  | -45%  | -46%          | 4.92                  |
|         | HUZ   | 187.44                                         | -36%                 | -37%  | -37%           | 1.39                                            | 208.99                | -27%  | -28%  | -29%          | 4.19                  |
|         | SCH   | 163.81                                         | -38%                 | -38%  | -39%           | 0.77                                            | 191.19                | -30%  | -31%  | -31%          | 1.18                  |
|         | China | 145.24                                         | -28%                 | -28%  | -28%           | 0.3                                             | 181.65                | -25%  | -25%  | -25%          | 0.9                   |

\* Δ-ADE represents the O3 response to ADE which is calculated from the difference between SimSF and SimBL.

---

## Author Response (AR1)

**Reply to comments from Referee #1 on "Impacts of aerosol direct effects on tropospheric ozone through changes in atmospheric dynamics and photolysis rates" by Xing et al.**

We thank the reviewer for the detailed and thoughtful review of our manuscript. Incorporation of the reviewer's suggestion has led to a much improved manuscript. Detailed below is our response to the issues raised by the reviewer. We also detail the specific changes incorporated in the revised manuscript in response to the reviewer's comments.

[Comment]: *The simulations appear to have been done carefully and the results are of some moderate interest, but the presentation is very difficult to wade through because of laborious analyses of model results that are of little interest and because of postage-stamp figures that seem like core dumps of obscure information.*
[Response]: We thank the reviewer for the suggestion. To improve the flow of the manuscript, we have reduced the amount of information presented and also moved some of the information into the supplementary material. Furthermore, we have increased the size of several figures to better present the information in the revised manuscript.

[Comment]: *it appears that the authors did not examine (or mention, unless I missed it) the aerosol effect on ozone through heterogeneous chemistry but I would expect this effect to be at least as large as the effects from dynamics and photolysis. Not accounting for the effect of aerosols on heterogeneous chemistry (for example through N2O5 hydrolysis) compromises in my opinion the policy-relevant conclusions about the sensitivity of ozone to aerosol reductions.*
[Response]: We agree with the reviewer that the heterogeneous reactions associated with aerosols have substantial impacts on ozone, including hydrolysis of $N_2O_5$, irreversible absorption of $NO_2$ and $NO_3$ as well as the uptake of $HO_2$, and are well documented in the literature (Tang et al., 2004; Tie et al., 2005; Liao and Seinfeld, 2005; Pozzoli et al., 2008; Li et al., 2011; Xu et al., 2012; Lou et al., 2014). Our model contains comprehensive treatment of heterogeneous hydrolysis of $N_2O_5$ (Davis et al., 2008; Sarwar et al., 2012; Sarwar et al., 2014). While our model accounts for such a heterogeneous reaction, we have not quantified its impacts on ozone in this study.
However, in this study, we focused our analysis on another important aspect of aerosol influence (ADE, the aerosol direct effects), i.e., scattering and absorption of incoming solar radiation and how the subsequent effects of the associated cooling suppresses atmospheric ventilation. The assessment of impacts of aerosol direct effects (ADE) is an important aspect of designing emission reduction strategies that seek co-benefits associated with reductions in both particulate matter and ozone. In this study, we examine the ADE impacts which were not well quantified in previous studies. It may be noted that all model calculations analyzed in this study included the heterogeneous $N_2O_5$ hydrolysis pathway.
We agree with the reviewer that all influences of aerosols on ozone need to be addressed before definite policy-relevant conclusions regarding their overall impact can be reached. China plans to implement stringent control actions aimed at lowering the ambient concentrations of $PM_{2.5}$ in the next two decades. It will be necessary to quantify all the possible influences resulting from this expected reduction of aerosols, including changes in heterogeneous reactions associated with aerosols, as well as the changes expected in ADE discussed in this manuscript. In addition, secondary aerosol and ozone comes from both NOx and VOCs emissions. Reducing aerosols by controlling gaseous precursors will also have substantial impacts on ozone (Xing et al., 2017), which needs further evaluation.

To address the reviewer's concern, we have clarified the scope of our analysis in the revised manuscript as below:

(Page 2 Line 21-25) "Many studies suggest that aerosols may have substantial impacts on ozone through heterogeneous reactions including hydrolysis of $N_2O_5$, irreversible absorption of $NO_2$ and $NO_3$, as well as the uptake of $HO_2$ (Tang et al., 2004; Tie et al., 2005; Li et al., 2011; Lou et al., 2014). While our model contains comprehensive treatment of the heterogeneous hydrolysis of $N_2O_5$ (Davis et al., 2008; Sarwar et al., 2012; Sarwar et al., 2014), we have not quantified its impacts on ozone in this study. However, ADE impacts on ozone have not been well evaluated previously."

(Page 10 Line 15-19) "Reducing aerosols will have substantial impacts on ozone. Quantification of the aerosol influence on ozone is important to understand co-benefits associated with reductions in both particulate matter and ozone. This study focused on the evaluation of ADE impacts which were not well quantified previously. However, the heterogeneous reactions associated with aerosols, as well as the impacts of emission controls of gaseous precursors on both aerosols and ozone also need to be studied in order to fully understand the influence from reducing aerosols on ambient ozone."

Reference

Li J., Wang Z., Wang X., Yamaji K., et al. Impacts of aerosols on summertime tropospheric photolysis frequencies and photochemistry over Central Eastern China. Atmospheric Environment, 2011, 45: 1817-1829.

Liao, H., Seinfeld, J.H., Global impacts of gas-phase chemistry–aerosol interactions on direct radiative forcing by anthropogenic aerosols and ozone, J. Geophys. Res., 2005, 110, D18208

Tang Y. H., Carmichael G. R., Kurata G., Uno I., et al. Impacts of dust on regional tropospheric chemistry during the ACE-Asia experiment: A model study with observations. Journal of Geophysical Research-Atmospheres, 2004, 109

Tie X. X., Madronich S., Walters S., et al. Assessment of the global impact of aerosols on tropospheric oxidants. Journal of Geophysical Research-Atmospheres 2005, 110

Lou S. J., Liao H., Zhu B. Impacts of aerosols on surface-layer ozone concentrations in China through heterogeneous reactions and changes in photolysis rates. Atmospheric Environment, 2014, 85: 123-138.

Pozzoli, L., Bey, I., Rast, S., Schultz, M.G., Stier, P., Feichter, J., Trace gas and aerosol interactions in the fully coupled model of aerosol-chemistry-climate ECHAM5-HAMMOZ: 1. Model description and insights from the spring 2001 TRACE-P experiment, J. Geophys. Res., 2008, 113, D07308

Xu, J., Zhang, Y.H., Zheng, S.Q., He, Y.J., Aerosol effects on ozone concentrations in Beijing: a model sensitivity study, J. Environ. Sci., 2012, 24 (4), 645–656

Davis, J. M., Bhave, P. V., and Foley, K. M.: Parameterization of N2O5 reaction probabilities on the surface of particles containing ammonium, sulfate, and nitrate, Atmos. Chem. Phys., 8, 5295–5311, doi:10.5194/acp-8-5295-2008, 2008.

Sarwar, G., Simon, H., Bhave, P., and Yarwood, G.: Examining the impact of heterogeneous nitryl chloride production on air quality across the United States, Atmospheric Chemistry & Physics, 12, 1-19, 2012.

Sarwar, G., Simon, H., Xing, J., Mathur, R.: Importance of tropospheric $ClNO_2$ chemistry across the Northern Hemisphere, Geophysical Research Letters, 41, 4050-4058, 2014.

Xing, J., Wang, S. X., Jang, C., et al. Overview of ABaCAS: an air pollution control cost-benefit and attainment assessment system and its applications in China, EM of Air & Waste Management Association, 2017 April.

[Comment]: *Page 2: what meteorological data assimilation is done in the WRF simulation and how would*

*it affect the sensitivity of dynamics to aerosols?*

[Response]:

We followed our previous coupled model design (Xing et al., 2015). The strength of nudging coefficients for four-dimensional data assimilation and indirect soil temperature nudging employed in WRF have been tested and chosen to improve model performance for meteorological variables without dampening the effects of radiative feedbacks. The nudging coefficient for both u/v-wind and potential temperature is set to 0.00005 $s^{-1}$, while 0.00001 $s^{-1}$ is used for nudging of the water vapor mixing ratio.

We have clarified it in the revised manuscript as below:

(Page 3 Line 7-12) "The meteorological inputs for WRF simulations were derived from the NCEP FNL (Final) Operational Global Analysis data which has 1 degree spatial and 6-hour temporal resolution. NCEP ADP Operational Global Surface Observations were used for surface reanalysis and four dimensional data assimilation. We have tested and chosen proper strength of nudging coefficients, i.e., 0.00005 $sec^{-1}$ is used for nudging of both u/v-wind and potential temperature, 0.00001 $sec^{-1}$ is used for nudging of water vapor mixing ratio, to improve model performance without dampening the effects of radiative feedbacks (Hogrefe et al., 2015; Xing et al., 2015)."

[Response]: We have reduced the content of these figures to focus on the most important aspects and added a colorbar legend for figures 5-6 in the revised manuscript.

[Comment]: *Page 6: I don't understand why dynamics decreases ozone deposition velocity in summer. During the daytime the ozone deposition velocity should be more limited by the surface resistance.*

[Response]: The dry deposition velocity is computed as the reciprocal of the sum for aerodynamic resistance (Ra), quasi-laminar resistance (Rb), and surface resistance (Rc). The Ra is a function of wind speed

and turbulence. Since the changes in dynamics decrease the wind speed, they thus increase Ra and consequently reduce dry deposition."

We have clarified this in the revised manuscript as below:

(Page 6 Line 23-25) "The stabilization of the atmosphere due to ΔDynamics leads to lower dry deposition rates (due to lower dry deposition velocity from the enhanced aerodynamic resistance) and thus increases surface $O_3$."

[Comment]: *Page 6: Page 9, line 1: summary states that aerosol effects improve the ozone simulation but I didn't see this demonstrated in the text.*

[Response]: Following the reviewer's suggestion, we have reduced the content of Figure 3 and summarized the comparison in Table 2 in the revised manuscript, as below:

(Page 5 Line 28- Page 6 Line 10)  "The impact of the ADE on $O_3$ is further explored by examining the relationship between the observed and simulated $O_3$ concentrations (DM1O$_3$, daily values of the cities located in China) as a function of the observed PM$_{2.5}$ concentrations (observed daily averaged values in those cities), as displayed in Figure 3. The predicted ozone concentrations under both low- and high- PM$_{2.5}$ levels are compared in Table 2. In regards to model performance for DM1O$_3$ simulations, the model generally exhibits a slight high bias in January but a low bias in July across the 5 regions. The inclusion of ADE moderately reduced $O_3$ concentrations in January and slightly increased $O_3$ in July, resulting in reduction in bias and improved performance for DM1O$_3$ simulation in both January and July for most of regions.

Interestingly, from low to moderate PM$_{2.5}$ levels (i.e., PM$_{2.5}$ < 120 μg m$^{-3}$), higher $O_3$ concentration occur with higher PM$_{2.5}$ concentrations, which is evident in both observations and simulations, suggestive of common precursors (e.g., NO$_x$), source sectors, and/or transport pathways contributing to both $O_3$ and PM$_{2.5}$ in these regions.  However, a negative correlation between $O_3$ and PM$_{2.5}$ is evident in winter when the PM$_{2.5}$ can reach high levels larger than 120 μg m$^{-3}$, indicating the strong ADE impacts on $O_3$ through both feedbacks to dynamics and photolysis which significantly reduced $O_3$."

[Figure]

**Black: Obs, Red: SimSF, Blue: SimNF, Green: SimBL**

January

[Figure]

**Figure 3: Observed and simulated surface O$_3$ concentration against PM$_{2.5}$ concentration (O$_3$ is daily 1h maxima of monitor sites over China, unit: μg m$^{-3}$; PM$_{2.5}$ is the daily average of those site, unit: μg m$^{-3}$)**

**Table 2: Comparison of model performance in ozone prediction across three simulations (monthly average of daily 1h maxima)**

| | Region | OBS (μg m$^{-3}$) | Low PM$_{2.5}$ (<60μg m$^{-3}$) Normalized Mean Bias | | | Δ-ADE (μg m$^{-3}$)* | OBS (μg m$^{-3}$) | High PM$_{2.5}$ (>60μg m$^{-3}$) Normalized Mean Bias | | | Δ-ADE (μg m$^{-3}$) |
| | | | SimSF | SimNF | SimBL | | | SimSF | SimNF | SimBL | |
|---|---|---|---|---|---|---|---|---|---|---|---|
| | JJJ | 62.52 | 3% | 4% | 5% | -1.05 | 37.02 | 22% | 36% | 53% | -11.36 |
| | YRD | 63.89 | 38% | 41% | 43% | -2.76 | 66.74 | 54% | 59% | 67% | -8.85 |
| January | PRD | 97.25 | 25% | 26% | 29% | -4.52 | 122.61 | 6% | 5% | 9% | -4.63 |
| | HUZ | 47.67 | 172% | 173% | 193% | -10.17 | 67.29 | 107% | 125% | 142% | -23.9 |
| | SCH | 88.63 | -43% | -40% | -38% | -3.85 | 111.19 | -5% | 2% | 8% | -13.78 |
| | China | 76.61 | 30% | 31% | 34% | -2.96 | 62.68 | 42% | 48% | 56% | -8.61 |
| | JJJ | 159.27 | -29% | -28% | -28% | -0.51 | 178.54 | -25% | -25% | -25% | 1.02 |
| | YRD | 171.04 | -31% | -31% | -32% | 0.84 | 233.13 | -24% | -25% | -23% | -0.51 |
| July | PRD | 129.02 | -20% | -19% | -20% | -0.09 | 312.21 | -44% | -45% | -46% | 4.92 |
| | HUZ | 187.44 | -36% | -37% | -37% | 1.39 | 208.99 | -27% | -28% | -29% | 4.19 |
| | SCH | 163.81 | -38% | -38% | -39% | 0.77 | 191.19 | -30% | -31% | -31% | 1.18 |
| | China | 145.24 | -28% | -28% | -28% | 0.3 | 181.65 | -25% | -25% | -25% | 0.9 |

* Δ-ADE represents the O$_3$ response to ADE which is calculated from the difference between SimSF and SimBL.

[Comment]: *Page 9: the summary presents as a punch line that one should decrease NOx emissions to improve both ozone and PM but this is hardly an original result.*

[Response]: Traditionally, the co-benefits from NOx controls for ozone and PM reductions were thought of as resulting from the fact that NOx acts as a common precursor for both O$_3$ and PM$_{2.5}$, thus a decrease of NOx emissions would be expected to reduce both ozone and PM. The analyses presented in this study reveals that the consideration of NOx controls is not only beneficial for directly reducing PM$_{2.5}$ and O$_3$, but also because of indirect benefits in reducing peak O$_3$ through the weakening of the ADE from the reduced PM$_{2.5}$, increasing the co-benefits from NO$_x$ controls for achieving both O$_3$ and PM$_{2.5}$ reductions.

To address the reviewer's concern, we have clarified this in the revised manuscript as below:

(Page 10 Line 11-14) "Traditionally, the co-benefits from NO$_x$ control for ozone and PM reduction are mostly thought of as resulting from the fact that NO$_x$ is a common precursor for both O$_3$ and PM$_{2.5}$. This study

suggests that effective controls on $NO_x$ will not only gain direct benefits for $O_3$ reduction, but also can indirectly reduce peak $O_3$ through weakening the ADE from the reduced $PM_{2.5}$, increasing the estimated co-benefits from $NO_x$ controls for achieving both $O_3$ and $PM_{2.5}$ reductions."

**Reply to comments from Referee #2 on "Impacts of aerosol direct effects on tropospheric ozone through changes in atmospheric dynamics and photolysis rates" by Xing et al.**

We thank the referee for a thoughtful and detailed review of our manuscript. Incorporation of the reviewer's suggestions has led to a much improved manuscript. Below we provide a point-by-point response to the reviewer's comments and summarize the changes that have been incorporated in the revised manuscript.

[Comment]: *In this paper, the authors have applied the WRF-CMAQ model to analyze the impact of aerosols on tropospheric ozone through their impacts on atmospheric dynamics and photolysis rates. Their results indicate that reducing aerosols may have negative impacts on ozone which need to be considered for air quality management in China. The topic is of general interest given the focus on reducing PM2.5 pollution. The simulations have been designed appropriately to address the goals of the study.*
[Response]: We thank the reviewer for the overall positive assessment of the manuscript and recognition of the implications of the results of the analysis presented.

[Comment]: *However, the authors have not considered the prominent way aerosols impact tropospheric ozone formation - via heterogeneous reactions - which leads me to question the conclusions of this study. Several studies have highlighted the role of aerosols in modulating ozone via heterogeneous reactions (eg., Liao and Seinfeld, 2005; Ti et al., 2005; Pozzoli et al., 2008; Xu et al., 2012; Lou et al., 2014), which have largely been ignored in this study. The authors need to provide a strong justification for ignoring the impact of aerosols on ozone via heterogeneous reactions, before I can recommend this paper for publication.*
[Response]: We agree with the reviewer that the heterogeneous reactions associated with aerosols have substantial impacts on ozone, including hydrolysis of $N_2O_5$, irreversible absorption of $NO_2$ and $NO_3$ as well as the uptake of $HO_2$, and are well documented in the literature (Tang et al., 2004; Tie et al., 2005; Liao and Seinfeld, 2005; Pozzoli et al., 2008; Li et al., 2011; Xu et al., 2012; Lou et al., 2014). Our model contains comprehensive treatment of heterogeneous hydrolysis of $N_2O_5$ (Davis et al., 2008; Sarwar et al., 2012; Sarwar et al., 2014). While our model accounts for such a heterogeneous reaction, we have not quantified its impacts on ozone in this study.

However, in this study, we focused our analysis on another important aspect of aerosol influence (ADE, the aerosol direct effects), i.e., scattering and absorption of incoming solar radiation and how the subsequent effects of the associated cooling suppresses atmospheric ventilation. The assessment of impacts of aerosol direct effects (ADE) is an important aspect of designing emission reduction strategies that seek co-benefits associated with reductions in both particulate matter and ozone. In this study, we examine the ADE impacts which were not well quantified in previous studies. It may be noted that all model calculations analyzed in this study included the heterogeneous $N_2O_5$ hydrolysis pathway.

We agree with the reviewer that all influences of aerosols on ozone need to be addressed before definite policy-relevant conclusions regarding their overall impact can be reached. China plans to implement stringent control actions aimed at lowering the ambient concentrations of $PM_{2.5}$ in the next two decades. It will be necessary to quantify all the possible influences resulting from this expected reduction of aerosols, including changes in heterogeneous reactions associated with aerosols, as well as the changes expected in ADE discussed in this manuscript. In addition, secondary aerosol and ozone comes from both NOx and VOCs emissions. Reducing aerosols by controlling gaseous precursors will also have substantial impacts on ozone (Xing et al., 2017), which needs further evaluation.

To address the reviewer's concern, we have clarified the scope of our analysis in the revised manuscript as below:

(Page 2 Line 21-25) "Many studies suggest that aerosols may have substantial impacts on ozone through heterogeneous reactions including hydrolysis of $N_2O_5$, irreversible absorption of $NO_2$ and $NO_3$, as well as the uptake of $HO_2$ (Tang et al., 2004; Tie et al., 2005; Li et al., 2011; Lou et al., 2014). While our model contains comprehensive treatment of the heterogeneous hydrolysis of $N_2O_5$ (Davis et al., 2008; Sarwar et al., 2012; Sarwar et al., 2014), we have not quantified its impacts on ozone in this study. However, ADE impacts on ozone have not been well evaluated previously."

(Page 10 Line 15-19) "Reducing aerosols will have substantial impacts on ozone. Quantification of the aerosol influence on ozone is important to understand  co-benefits associated with reductions in both particulate matter and ozone. This study focused on the evaluation of ADE impacts which were not well quantified previously. However, the heterogeneous reactions associated with aerosols, as well as the impacts of emission controls of gaseous precursors on both aerosols and ozone also need to be studied in order to fully understand the influence from reducing aerosols on ambient ozone."

[Comment]: *P4L20: Please clarify if this is for observations. The observations are hardly visible in Figure 2.*
[Response]: Yes, this is for observations. We have replotted Figure 2 in the revised manuscript to make the comparison more evident, as below
(Page 4 Line 28 - Page 5 Line 3) "In January, higher $DM1O_3$ concentrations are seen in PRD where solar radiation is stronger than in the north. The model generally captured the spatial pattern with highest $DM1O_3$ in PRD over the simulated domain. Simulated DM1O3 in YRD, SCH and HUZ are higher than observations. Such overestimation might be associated with the relative coarse spatial resolution in the model. NO titration effects in urban areas were not well represented in the model."

[Figure]

**Figure 2: Observed and simulated O₃ and its response to ADE (monthly average of daily 1h maxima, μg m⁻³)**

[Comment]: *P4L24: Please provide a reference for "In January, O3 production in north China is VOC-limited regime"*

[Response]: Following reviewer's suggestion, a reference has been added as below:

(Page 5 Line 6-8) "In January, $O_3$ production in north China is occurring in a VOC-limited regime (e.g., Liu et al., 2010), thus increases in NOx at the surface stemming from the stabilized atmosphere by ADE inhibit $O_3$ formation due to enhanced titration by NO."

Liu, X.H., Zhang, Y., Xing, J., Zhang, Q., Wang, K., Streets, D.G., Jang, C., Wang, W.X. and Hao, J.M., 2010. Understanding of regional air pollution over China using CMAQ, part II. Process analysis and sensitivity of ozone and particulate matter to precursor emissions. Atmospheric Environment, 44(30), pp.3719-3727.

[Comment]: *P4: It would also be helpful to see maps of PM2.5 to assess if the reductions in O3 due to aerosol feedbacks are co-located with PM concentrations.*

[Response]: Following the reviewer's suggestion, we made the plot of $PM_{2.5}$ concentrations as Figure R1

[Figure]

Figure R1 spatial distribution of $PM_{2.5}$ concentration and $O_3$ response to ADE

It clearly shows that the ADE effects on $O_3$ are co-located with PM concentrations. Besides, we calculated the $O_3$ responses to ADE under low- and high- $PM_{2.5}$ levels, summarized in Table 2. Mostly, the $O_3$ responses to ADE are larger under high $PM_{2.5}$ levels, indicating the positive correlations between $O_3$ responses and PM levels.

**Table 2: Comparison of model performance in ozone prediction across three simulations (monthly average of daily 1h maxima)**

| | Region | OBS ($\mu g\ m^{-3}$) | Low PM$_{2.5}$ (<60$\mu g\ m^{-3}$) Normalized Mean Bias | | | Δ-ADE ($\mu g\ m^{-3}$)* | OBS ($\mu g\ m^{-3}$) | High PM$_{2.5}$ (>60$\mu g\ m^{-3}$) Normalized Mean Bias | | | Δ-ADE ($\mu g\ m^{-3}$) |
| | | | SimSF | SimNF | SimBL | | | SimSF | SimNF | SimBL | |
|---|---|---|---|---|---|---|---|---|---|---|---|
| January | JJJ | 62.52 | 3% | 4% | 5% | -1.05 | 37.02 | 22% | 36% | 53% | -11.36 |
| | YRD | 63.89 | 38% | 41% | 43% | -2.76 | 66.74 | 54% | 59% | 67% | -8.85 |
| | PRD | 97.25 | 25% | 26% | 29% | -4.52 | 122.61 | 6% | 5% | 9% | -4.63 |
| | HUZ | 47.67 | 172% | 173% | 193% | -10.17 | 67.29 | 107% | 125% | 142% | -23.9 |
| | SCH | 88.63 | -43% | -40% | -38% | -3.85 | 111.19 | -5% | 2% | 8% | -13.78 |
| | China | 76.61 | 30% | 31% | 34% | -2.96 | 62.68 | 42% | 48% | 56% | -8.61 |
| July | JJJ | 159.27 | -29% | -28% | -28% | -0.51 | 178.54 | -25% | -25% | -25% | 1.02 |
| | YRD | 171.04 | -31% | -31% | -32% | 0.84 | 233.13 | -24% | -25% | -23% | -0.51 |
| | PRD | 129.02 | -20% | -19% | -20% | -0.09 | 312.21 | -44% | -45% | -46% | 4.92 |
| | HUZ | 187.44 | -36% | -37% | -37% | 1.39 | 208.99 | -27% | -28% | -29% | 4.19 |
| | SCH | 163.81 | -38% | -38% | -39% | 0.77 | 191.19 | -30% | -31% | -31% | 1.18 |
| | China | 145.24 | -28% | -28% | -28% | 0.3 | 181.65 | -25% | -25% | -25% | 0.9 |

**\*** Δ-ADE represents the O$_3$ response to ADE which is calculated from the difference between SimSF and SimBL.

We have added corresponding discussions to the revised manuscript, as below:

(Page 6 Line 3-5) "Comparing the O$_3$ responses to ADE (see Δ-ADE in Table 2) under low- and high- PM$_{2.5}$ levels, reveals that O$_3$ responses to ADE are larger under high PM$_{2.5}$ levels, indicating the positive correlations between O$_3$ responses and PM$_{2.5}$ levels."

[Comment]: *P4-5: How significant are the changes in O3 in response to Dynamics and Photolysis*

[Response]: Out results suggest that ADE reduced surface daily maxima 1h O$_3$ (DM1O3) in China by up to 39 $\mu g\ m^{-3}$ through the combination of ΔDynamics and ΔPhotolysis in January, but enhanced surface DM1O3 by up to 4 $\mu g\ m^{-3}$ in July.

The enhancement in peak O$_3$ in summer found in this study is different from the traditional expectation of a negative O$_3$ response to ADE when solely considering the reduction in solar radiance.

[revised manuscript text omitted]

* $\Delta$-ADE represents the O$_3$ response to ADE which is calculated from the difference between SimSF and SimBL.

[Figure]

**Figure 1: Simulation domain and locations of 5 selected regions in China. Note: JJJ=Jing-Jin-Ji area, YRD=Yangzi-River-Delta area, PRD=Perl-River-Delta area, SCH=Sichuan Basin area, HUZ=Hubei-Hunan area.**

[Figure]

**Figure 2: Observed and simulated O₃ and its response to ADE (monthly average of daily 1h maxima, μg m⁻³)**

[Figure]

**Figure 3: Observed and simulated surface O₃ concentration against PM₂.₅ concentration (O₃ is daily 1h maxima of monitor sites over China, unit: μg m⁻³; PM₂.₅ is the daily average of those site, unit: μg m⁻³)**

[Figure]

**Figure 4: Diurnal variation of selected integrated process contributions to surface O₃ concentration in JJJ (The calculation is based on the average of grid cells in JJJ; a. Baseline is the simulated O₃ in SimBL, unit: ppb hr⁻¹; b. Δ-ADE is the difference in normalized IPRs between simulations, unit: hr⁻¹: delta_Dynamic is the difference between SimSF and SimNF, delta_Photolysis is the difference between SimNF and SimBL, delta_Total is the difference between SimSF and SimBL)**

[Figure]

**Figure 5: Vertical profile of integrated process contributions to surface O₃ concentration at noon in JJJ (full-layer heights above ground are 40, 96, 160, 241, 355, 503, 688, 884, 1100, 1357m; a. Baseline is the simulated O₃ in SimBL, unit: ppb hr⁻¹; b. ΔDynamic is the difference in normalized IPRs between SimSF and SimNF, unit: hr⁻¹; d. ΔPhotolysis is the difference in normalized IPRs between SimNF and SimBL, unit: hr⁻¹; c. ΔTotal is the difference in normalized IPRs between SimSF and SimBL, unit: hr⁻¹)**

[Figure]

**Figure 6: Integrated process contributions to daytime near-ground-level O₃ under different PM₂.₅ level in JJJ (between the ground and 350m AGL, model layer 1-5)**

[Figure]

**Figure 7: Impacts of ADE on surface O$_x$ and OH (monthly average of noon time 11am-1pm local time)**

[Figure]

[Figure]

**Figure 8: Vertical profile of integrated reaction rates in JJJ at noon (full-layer heights above ground are 40, 96, 160, 241, 355, 503, 688, 884, 1100, 1357m; Baseline is the simulation in SimBL; ΔDynamic is the difference between SimSF and SimNF; ΔPhotolysis is the difference between SimNF and SimBL; ΔTotal is the difference between SimSF and SimBL; $P_{totalO_x}$ is total $O_x$ production rate, unit: ppb hr$^{-1}$; OH CL is OH chain length; $P_{NewOH}$ is the production rate of new OH, unit: ppb hr$^{-1}$; $P_{ReactedOH}$ is the production rate of reacted OH, unit: ppb hr$^{-1}$; $P_{H2O2}$ is the production rate of H$_2$O$_2$, unit: ppb hr$^{-1}$; $P_{HNO3}$ is the production rate of HNO$_3$, unit: ppb hr$^{-1}$; the ratio of $P_{H2O2}/P_{HNO3}$ is only shown for layer 1-5)**